# MODEL FUSION VIA NEURON INTERPOLATION

## ABSTRACT

Model fusion aims to combine the knowledge of multiple models by creating one representative model that captures the strengths of all of its parents. However, this process is non-trivial due to differences in internal representations, which can stem from permutation invariance, random initialization, or differently distributed training data. We present a novel, neuron-centric family of model fusion algorithms designed to integrate multiple trained neural networks into a single network effectively regardless of training data distribution. Our algorithms group intermediate neurons of parent models to create target representations that the fused model approximates with its corresponding sub-network. Unlike prior approaches, our approach incorporates neuron attribution scores into the fusion process. Furthermore, our algorithms can generalize to arbitrary layer types. Experimental results on various benchmark datasets demonstrate that our algorithms consistently outperform previous fusion techniques, particularly in zero-shot and non-IID fusion scenarios. We make our code publically available.

## 1 INTRODUCTION

As modern Deep Neural Networks (DNNs) continue to grow in scale, retraining them on new data is often prohibitively expensive or infeasible, especially in settings where data privacy must be preserved. Model fusion offers an appealing alternative: instead of retraining, one may combine *independently trained* models directly. Two influential contributions in this area are OTFusion (Singh and Jaggi, 2020), and Git Re-Basin (Ainsworth et al., 2022). This line of work has been motivated in part by the Linear Mode Connectivity (LMC) conjecture, which posits that independently trained networks can be connected by low-loss paths (Frankle et al., 2020). While the empirical evidence for LMC is strong (Theus et al., 2025), prior studies reveal that even models trained on identical data may learn divergent internal features (Li et al., 2015), undermining the premise of weight/activation matching. While (Ainsworth et al., 2022) argue that barriers vanish in sufficiently large networks, their own experiments highlight numerous failure modes, including simple architectures such as MLPs on MNIST trained with SGD and a low learning rate.

In this work, we identify three key gaps in prior research on model fusion, which we address. **Reproducibility.** Many open-source implementations re-implement the same algorithm separately for each architecture, limiting generality and making systematic benchmarking difficult. **Base model quality.** Prior studies often report results using base models with accuracies below the standard typically achieved by the same architectures. While the aim of those works was not to train state-of-the-art baselines, this raises the question of whether fusion methods could show similar improvements if the base models themselves were trained more thoroughly; even with simple techniques such as using CutMix (Yun et al., 2019). **Heterogeneous data.** Experiments on models trained with different data distribution settings remain limited, with some works focusing on surrogate metrics such as loss or calibration rather than accuracy. In Section 5, we show empirically that existing methods struggle in zero-shot fusion under such heterogeneous conditions, which typically arise in the context of Federated Learning (FL).

**Contributions.** Motivated by these shortcomings, we introduce a family of neuron-centric fusion algorithms with the following key innovations: **(a) Casting fusion as a principled representation-matching problem**, yielding a two-stage algorithm that performs well on various data settings. **(b) Incorporating neuron saliency into alignment**, improving performance across our methods and enhancing existing approaches such as Git Re-Basin and Transformer OTFusion. **(c) A flexible open source re-implementation of existing algorithms**, to allow for benchmarking in the future.

Table 1: Comparison of Algorithm Features

| Algorithm | Linear layers | Trans-formers | Any differe-entiable arch. | Can fuse > 2 models | Can fuse diff. widths | Can fuse diff. depths | Gains from imp. scores | High zero-shot acc. |
|---|---|---|---|---|---|---|---|---|
| OTFusion | ✓ | ✓ | ✗ | ✓ | ✓ | ✗ | ✗ | ✗ |
| Git-Rebasin | ✓ | ✗ | ✗ | ✗ | ✗ | ✗ | ✓ | ✗ |
| HF (Ours) | ✓ | ✓ | ✓ | ✗ | ✗ | ✓ | ✓ | ✓ |
| KF (Ours) | ✓ | ✓ | ✓ | ✓ | ✓ | ✓ | ✓ | ✓ |

## 2 RELATED WORK

### 2.1 FUSION ALGORITHMS

**OTFusion** (Singh and Jaggi, 2020) formulates neuron alignment as a discrete optimal transport (OT) problem. Given multiple models, OTFusion selects an initial reference model, aligns each of the others' layers to its layers via optimal transport on neuron activations, and averages the aligned parameters to produce a fused model. One downside of OTFusion is that it was initially designed to handle only linear layers. Later work (Imfeld et al., 2023) adapted OTFusion to the transformer (Vaswani et al., 2017) architecture. However, it doesn't work out-of-the-box with any architecture.

**Git Re-Basin** (Ainsworth et al., 2022) proposes three strategies to perform neuron alignment. In this work, we focus exclusively on the "Matching Activations" approach, which is most directly comparable to our methods. Activation-based Git Re-Basin finds a permutation matrix that minimizes the L2 distance between neuron activations across models. This makes it equivalent to OTFusion, when the latter is made to use an activations-based ground metric. While effective, activations-based Git Re-Basin is limited to pairwise fusion, restricts itself to a one-to-one matching paradigm, and does not account for neuron saliency. We discuss our extension to incorporate scores in Appendix C. Subsequent work (Jordan et al., 2022) investigates the factors contributing to Git Re-Basin's limited zero-shot accuracy and proposes a remedy based on rescaling the weights of the fused model.

**Federated Learning algorithms**. In federated learning (FL), decentralized clients train local models on private data and periodically send updates to a central server, which aggregates them into a global model. Thus, model fusion on non-IID data distributions is central to FL.

Some of the most well-known methods in this domain include **Federated Averaging (FedAvg)** (McMahan et al., 2017) and **Federated Matching Averaging (FedMA)** (Wang et al., 2020). The former averages model weights across clients proportionally to the number of local updates or data samples. While simple and popular, FedAvg performs poorly on independently-trained models, as weights can diverge significantly in the presence of heterogeneous data, especially for deeper architectures. FedMA aims to address this challenge by aligning neurons before averaging. However, FedMA requires retraining the fused model after the alignment of every layer to ensure performance recovery, which makes it a *non-zero-shot* fusion algorithm.

Lastly, we briefly mention the traditional methods of aggregating knowledge from different models. **Ensembles**, which average the predictions of base models, typically, represent an upper bound on the performance we can achieve by zero-shot fusion, however, *at the expense of computational overhead*. **Vanilla Averaging** blindly averages the weights of two identical models without alignment. In **Knowledge Distillation (KD)** (Hinton et al., 2015), a model is trained to predict soft-targets originating another model. While KD was initially developed for model compression, later work extended it for the multi-teacher setting (Asif et al., 2019).

### 2.2 NEURON ATTRIBUTION

A novel feature of our work is incorporating the optional use of neuron attribution scores, commonly referred to as *neuron importance scores*, into the fusion process to bias the preservation of salient

features. **Uniform** importance distributes an equal weight of $1/n$ on each neuron in a layer of $n$ neurons. **Conductance** (Dhamdhere et al., 2018) extends *Integrated Gradients* (Sundararajan et al., 2017), which attributes feature importance by integrating gradients along a straight-line path from a baseline input to the actual input. Conductance applies the chain rule to propagate these importance scores to hidden neurons, enabling internal saliency estimation. **DeepLIFT** (Shrikumar et al., 2017) provides another method for attributing importance scores to neurons. It computes the contribution of each neuron by comparing its activation to a reference activation and propagating these differences through a modified chain rule. Unlike gradient-based methods, DeepLIFT can assign non-zero importance scores even when gradients are zero or poorly behaved and requires only a single backward pass, making it computationally efficient.

## 3 MOTIVATION

In a neural network, each layer of neurons can be interpreted as encoding a specific amount of information which is used by future layers to produce an output (Here, a "neuron" denotes a set of activations controlled by a common set of weights — this corresponds to a channel for a convolutional layer or a single embedding dimension for a transformer). Consequently, for the purpose of model fusion, a natural goal is to preserve the information contained in the neurons of the base models by ensuring that each base model neuron is closely represented by a neuron in the fused model. One obvious metric for neuron closeness is the squared L2 or Euclidean distance between the (pre)activations, which has been already been used in the context of model fusion by Singh and Jaggi (2020) and Ainsworth et al. (2022). This will motivate our definition of **representation cost** for a given layer of the fused model. An extension to the raw squared L2 distance is weighing them by *neuron importance scores* which intuitively penalizes misrepresenting more important neurons.

We now introduce the notation. A (Deep) Neural Network (DNN) can be viewed as a function $f_{\mathbf{w}} : \mathbb{R}^d \mapsto \mathbb{R}^o$ parameterized by weights $\mathbf{w}$ where $d$ is the number of input features and $o$ is the number of output features. For many model architectures, $f_{\mathbf{w}}$ can be decomposed into $L$ sequential functions $f_{\mathbf{w}} = f^L_{\mathbf{w}_L} \circ \cdots \circ f^1_{\mathbf{w}_1}$ with $L$ corresponding to the depth of the model. Furthermore, it is possible to *arbitrarily* group those functions into so-called **levels**. For example, we can decompose $f_{\mathbf{w}} = f^3_{\mathbf{w}_3} \circ f^2_{\mathbf{w}_2} \circ f^1_{\mathbf{w}_1}$ into $f_{\mathbf{w}} = \widehat{f^2}_{\widehat{\mathbf{w}}_2} \circ \widehat{f^1}_{\widehat{\mathbf{w}}_1}$, where $\widehat{f^2}_{\widehat{\mathbf{w}}_2} = f^3_{\mathbf{w}_3}$ and $\widehat{f^1}_{\widehat{\mathbf{w}}_1} = f^2_{\mathbf{w}_2} \circ f^1_{\mathbf{w}_1}$ are individual **levels**. An important observation is that layers with branching (e.g. skip connections) can be contained in a single level so that the functions may be composed sequentially. In our algorithms, we will fix the weights of each level sequentially from the first level to the last.

Now, for fusion, we let $\mathcal{M} = \{M_1, M_2, \ldots, M_n\}$ be a collection of pretrained base models. Each model is strategically partitioned to have $L$ levels. To keep the notation simple, we will define the **representation cost** for a fixed level $l$, where we assume the weights of the fused model for all previous layers have already been fixed. We let $\mathbf{z}^{M_k}$ be the output vector of model $M_k$ at this level. We denote by $\mathbf{z} = \mathrm{concat}(\mathbf{z}^{M_1}, \ldots, \mathbf{z}^{M_n}) \in \mathbb{R}^{d^{\mathcal{M}}}$ the concatenated outputs, of total size $d^{\mathcal{M}}$. For a fused model $\mathcal{F}$ with weights $\mathbf{w}$ at level $l$, we write $\mathbf{z}^{\mathcal{F}_{\mathbf{w}}} \in \mathbb{R}^{d^{\mathcal{F}}}$ for its outputs with size $d^{\mathcal{F}}$. These outputs are in most cases the **activations** or **preactivations** at a given level. We also use $s_j$ for the importance score of neuron $j$ (of the concatenated outputs $\mathbf{z}$). Now, we define the **representation cost** of using weights $\mathbf{w}$ at level $l$ of the fused model $\mathcal{F}$ (for a given input $\mathbf{x}$):

$$J_{\mathbf{w}}(\mathbf{x}) \ = \ \sum_{j=1}^{d^{\mathcal{M}}} s_j \left( \min_{k \in \{1, \ldots, d^{\mathcal{F}}\}} \left\{ \left( z_k^{\mathcal{F}_{\mathbf{w}}}(\mathbf{x}) - z_j(\mathbf{x}) \right)^2 \right\} \right) \tag{1}$$

Intuitively, for each neuron in the concatenated base outputs $\mathbf{z}(\mathbf{x})$, we compute the squared L2 distance to **its closest neuron in the fused model output** $\mathbf{z}^{\mathcal{F}_{\mathbf{w}}}(\mathbf{x})$, which can be viewed as its **representative neuron**, and sum these distances. To solve for the desired weights $\mathbf{w}$ of the fused model $\mathcal{F}$ at level $l$, we would in principle choose $\mathbf{w}$ to minimize this cost.

Popular layer-wise fusion algorithms (Singh and Jaggi, 2020; Ainsworth et al., 2022) similarly optimize the L2 distance to obtain soft/hard permutation matrices for neuron alignment and proceed to average the aligned weights of the base models, layer-by-layer. In Section 5 we empirically show that these algorithms: **a**) fail to perform on par with base models in zero-shot fusion, i.e. they require a fine-tuning phase; and **b**) fail to achieve meaningful transfer knowledge in the non-IID regime.

We hypothesize that these shortcomings arise because: **a**) current algorithms ignore how the fused model evolves as the algorithm iterates through the levels, treating each level in isolation by only tracking past permutation or alignment matrices, without accounting for potential changes content of previous level outputs caused by the adjustment of weights; and **b**) not all neurons contribute equally to a model's prediction on average, but are getting averaged with equal importance. This can especially be an issue in the non-IID or sharded settings, where the activations on unseen data may be noisy or irrelevant, or the weights learn to extract different features.

## 4    PROPOSED METHOD

To optimize the objective in Eq. (1), we decouple the objective by introducing an auxiliary vector $\mathbf{T}$ of size $d^{\mathcal{M}}$, which we refer to as the **target vector** and it enables a more tractable decomposition of the cost function. This yields the following Theorem.

**Theorem 1.** *Let $\mathbf{T} \in \mathbb{R}^{d^{\mathcal{M}}}$ be a vector whose components are the importance-weighted means of clustered outputs. Then the representation cost can be decomposed as follows:*

$$J_{\mathbf{w}}(\mathbf{x}) \;=\; \underbrace{\sum_{k=1}^{d^{\mathcal{F}}} \sum_{j \in R_k} s_j \left(z_k^{\mathcal{F}}(\mathbf{x}) - T_k\right)^2}_{\text{approximation error}} + \underbrace{\sum_{k=1}^{d^{\mathcal{F}}} \sum_{j \in R_k} s_j \left(T_k - z_j(\mathbf{x})\right)^2}_{\text{grouping error}} \qquad (2)$$

*where $R_k$ is the set (or cluster) of base model neurons that fused model neuron $k$ represents in the minimum cost assignment in Eq. (1).*

*Proof.* See Appendix A. □

We observe that the sum of $J_{\mathbf{w}}(\mathbf{x})$ over a batch is subdifferentiable with respect to $\mathbf{w}$ and that this objective could potentially be optimized with subgradient descent in the same spirit as the weighted K-means objective (Bottou and Bengio, 1994). However, we leave this for future work.

The resulting objective naturally decomposes into two interpretable components. The **grouping error** measures how well the original neurons cluster together – specifically, how far each output $z_j$ is from the importance-weighted cluster center $T_k$ it was assigned to. The **approximation error**, on the other hand, quantifies how closely the fused model can reproduce these cluster centers through its own output neurons $\mathbf{z}$.

For a batch of $B$ values of $\mathbf{x}$, minimizing the total grouping error by constructing an optimal $\mathbf{T}$ through an effective clustering of the layer outputs $z_j$ is a critical challenge. This problem corresponds to the K-means problem in $\mathbb{R}^B$ which is known to be NP-hard in general (Aloise et al., 2009). Nonetheless, practical approximation algorithms such as Lloyd's algorithm (Lloyd, 1982) or local search-based methods (Kanungo et al., 2002) can be employed to obtain effective solutions in practice.

After having determined $\mathbf{T}$ (and hence, the clusters $R_k$), we can solve for the weights of a level by minimizing the approximation error, which is equivalent to a weighted mean squared error loss function. We can choose to either keep the weights of previous level frozen (and just optimize the current level), or optimize the whole subnetwork. In this work, we choose the former:

$$\mathbf{w}^* \;=\; \arg\min_{\mathbf{w}} \; \mathbb{E}_{\mathbf{x} \sim D} \left[ \sum_{k=1}^{d^{\mathcal{M}}} \sum_{j \in R_k} s_j \left(z_k(\mathbf{x}) - T_k(\mathbf{x})\right)^2 \right] \qquad (3)$$

This decomposition offers a more interpretable and stable optimization target by isolating the challenges of clustering and function fitting, instead of trying to solve them jointly.

### 4.1    PROPOSED ALGORITHM

Following the derivation in Theorem 1, we propose a two-step algorithm to find weights $\mathbf{w}$ that minimize Eq. (2) in expectation. The algorithm constructs the fused model in a bottom-up way, iterating through the levels of base models, and producing the corresponding level of the fused model.

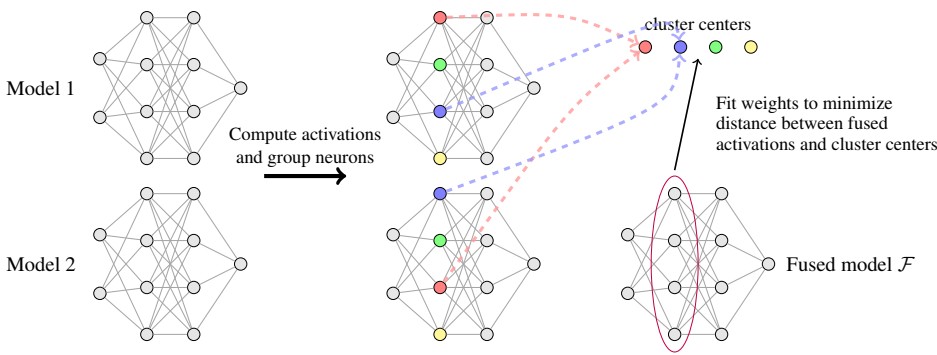

Figure 1: Overview of our method. Given two MLPs with two hidden layers of size 4 with each layer defined as its own level, we compute level outputs for the base models, cluster their neurons, and train the first level of the fused model to match the cluster centers using MSE loss. The process is repeated in subsequent levels.

At each level, the algorithm computes a matching/clustering to minimize the grouping error. It then uses this clustering to compute cluster centers (weighted by importance score). Finally, it uses the cluster centers as a target for the current level's outputs. We then fit weights of the fused model to minimize the approximation error. In practice, we use a finite batch to approximate the expected representation cost. An intuitive illustration of our algorithm can be found in Fig. 1. High-level pseudo-code is provided in Algorithm 1. For more implementation details, refer to Appendix F.

---

**Algorithm 1** Neuron Interpolation Model Fusion

---

**Require:** Trained base models $\mathcal{M} = \{M_k\}_{k=1}^n$, neuron importance scores for each base model $\{s_j^{M_1,l}, \ldots, s_j^{M_k,l}\}_{l=1}^L$, fusion dataset $\mathbf{X} \in \mathbb{R}^{B \times d}$
**Ensure:** Fused model $\mathcal{F}$ with weights $W_{\mathcal{F}}$
1: **for** each level $l = 1, 2, \ldots, L$ **do**
2:     Gather level $l$ outputs: $\mathbf{z} = \text{concat}\left(\mathbf{z}^{M_1,l}, \ldots, \mathbf{z}^{M_K,l}\right)$
3:     Gather level $l$ scores: $\mathbf{s} = \text{concat}\left(\mathbf{s}^{M_1,l}, \ldots, \mathbf{s}^{M_K,l}\right)$
4:     Obtain the clusters $(R_k)_{k=1}^{d^{\mathcal{F}}}$ for every output $z_j$         {Hungarian Matching or K-means}
5:     **for** each centroid $k = 1, \ldots, d^{\mathcal{F}}$ **do**
6:        $T_k \leftarrow \frac{\sum_{j \in R_k} s_j z_j}{\sum_{j \in R_k} s_j}$                      {Compute importance-weighted mean}
7:     **end for**
8:     Optimize the weights $\mathbf{w}$ of the current level of $\mathcal{F}$ using:

$$\mathbf{w} \leftarrow \arg\min_{\mathbf{w}} \sum_{m=1}^{B} \left[\sum_{k=1}^{d^{\mathcal{M}}} \sum_{j \in R_k} s_j \left(z_k^{\mathcal{F}}(\mathbf{x}_m) - T_k(\mathbf{x}_m)\right)^2\right]$$

{Fit Fused Model activations to cluster centers}

9: **end for**
10: **return** Fused Model $\mathcal{F}$

---

## 4.2 MINIMIZING THE GROUPING AND APPROXIMATION ERRORS

### 4.2.1 GROUPING ERROR

For the grouping error, we distinguish between two cases based on the architecture of the base models and the constraints imposed on the assignment. More details can be found in Appendix B.1.

(a) *Equal-size models with level-wise one-to-one matching.* This case induces a cost that can be minimized using the Hungarian Matching algorithm (Kuhn, 1955) as discussed in Section 3. We refer to this special case of our algorithm as **Hungarian Fusion (HF)**.

(b) *General case with arbitrary model sizes.* In this setup, the grouping problem becomes a general clustering task. As highlighted in Section 4.1, a solution can be approximated using heuristic K-means algorithms. We refer to this general case as **K-means Fusion (KF)**.

### 4.2.2 APPROXIMATION ERROR

For the approximation error, we distinguish two cases. More details can be found in Appendix B.2.

(a) *Linear levels.* When all trainable levels are affine transformations such as fully connected or convolutional layers, the outputs $z_j$ are linear functions of the level weights $\mathbf{w}$, and the problem becomes weighted least squares, which has a closed-form solution. In this case, we project the base models' level outputs onto the image of the previous fused level's outputs, before running HF or KF for the level. We call these algorithms the **Linear** version of HF and KF respectively.

(b) *General case.* For arbitrary differentiable (and possibly non-linear) levels, we optimize Eq. (3) using stochastic gradient descent (SGD). In this setting, initialization plays a critical role. A simple strategy is to initialize with the weights of the corresponding level from one of the base models, perturbed by noise $\epsilon$. Note that for linear levels, the objective is convex, reducing to case (a). For this case, we will only consider KF and refer to this algorithm as the **Gradient** version of KF.

### 4.2.3 GUARANTEES

We now present theoretical guarantees for Algorithm 1 under specific conditions.

**Theorem 2.** *Let the parametrized levels of the base models and fused model be affine functions. Then:*

*(a) For two models with equal-sized levels and a one-to-one matching constraint, the **Hungarian Fusion** algorithm returns an optimal solution to the decoupled objective in Eq. (2).*

*(b) For an arbitrary number of models with possibly different numbers of neurons per level, the **K-means Fusion** algorithm produces a solution whose representation cost is at most $(9 + \epsilon)$ times the optimal, when using the local-search algorithm from Kanungo et al. (2002).*

*Proof.* See Appendix A. □

## 5 EXPERIMENTS

We evaluate our fusion algorithms across three distinct training regimes, each characterized by a different data distribution used to train the base models. This setup is designed to test the robustness and generality of our method. We benchmark our algorithms against previous baselines, ensembles, vanilla averaging, KD and Linear Probing (LP). We note that both KD and LP are special cases of our algorithm, where in the former we treat the whole model as a single level, and in the latter we skip all layers except the classifier head.

### 5.1 ON THE PERFORMANCE OF BASE MODELS IN NON-IID SETUPS

Before presenting our results, we emphasize an important consideration in evaluating base model performance under non-IID conditions. In these settings, each model has access to only a small and often imbalanced portion of the dataset, which naturally limits its accuracy. For example, in 6-way or 8-way splits, each model sees only 10-20% of the full data, leading to lower performance compared to centralized training.

Despite these constraints, gains in this setup are meaningful. Improving over weak, heterogeneous base models in a zero-shot setting is a challenging task, and our method demonstrates robustness where baseline methods fail.

### 5.2 SHARDED SETUP

We train base models on "sharded" data splits, which represent an extreme non-IID case where each model sees all the samples from different classes. This leads the base models to class-specific overfitting and learning diverse representations. This setup is typically considered in FL research,

where it serves as a stress test due to it's extremity. We further assume that the fusion dataset is skewed and drawn from one of the base models, mimicking the FL constraint that data cannot be shared across servers due to privacy and communication costs. If models had access to the entire dataset, training directly on it would be more effective than model fusion. Details of the partitioning procedure are provided in Appendix D.

We evaluate all fusion algorithms in a *zero-shot* setting, where models are fused without further retraining. This reflects the above assumption that the fusion dataset is skewed, and therefore, retraining on it would not result in improved performance compared to the corresponding base model. We present some results for ViTs on CIFAR-100 in Table 2. Results for Tiny-ImageNet can be found in Appendix G.2, and results for VGGs on CIFAR-10 can be found in Appendix G.1.

Table 2: **Test accuracy** comparison when fusing ViT networks on CIFAR-100 for **Sharded** splits. For the table with full details, please refer to Table 15.

| Method | 2-WAY SPLIT | 4-WAY SPLIT | | 6-WAY SPLIT | | |
|---|---|---|---|---|---|---|
| Individual Models | $38.6_{\pm 0.5}$ $37.2_{\pm 0.6}$ | $20.4_{\pm 0.2}$, $19.5_{\pm 0.2}$ | $19.9_{\pm 0.1}$ $19.2_{\pm 0.3}$ | $14.3_{\pm 0.2}$, $13.2_{\pm 0.2}$ | $13.7_{\pm 0.3}$ $12.8_{\pm 0.3}$ | $13.5_{\pm 0.2}$ $12.2_{\pm 0.6}$ |
| Ensemble | $63.7_{\pm 0.4}$ | $53.4_{\pm 1.8}$ | | $45.4_{\pm 2.0}$ | | |
| Vanilla Averaging | $2.2_{\pm 0.6}$ | $1.4_{\pm 0.2}$ | | $1.1_{\pm 0.3}$ | | |
| KD | $50.4_{\pm 1.4}$ | $40.3_{\pm 0.9}$ | | $34.3_{\pm 1.1}$ | | |
| LP | $51.8_{\pm 0.6}$ | $37.1_{\pm 0.8}$ | | $28.0_{\pm 0.8}$ | | |
| Transf. OTF acts | $2.3_{\pm 0.6}$ | $1.0_{\pm 0.0}$ | | $1.0_{\pm 0.0}$ | | |
| Transf. OTF wts | $4.4_{\pm 1.3}$ | $1.5_{\pm 0.3}$ | | $1.2_{\pm 0.3}$ | | |
| HF Gradient (Ours) | $\mathbf{55.5}_{\pm 0.8}$ | - | | - | | |
| KF Gradient (Ours) | $54.7_{\pm 1.2}$ | $\mathbf{43.5}_{\pm 0.5}$ | | $\mathbf{37.4}_{\pm 0.8}$ | | |

We additionally present an experiment motivated by a potential real-world scenario performed on the BloodMNIST dataset (Yang et al., 2023). BloodMNIST contains 17,092 images of blood cells divided into 8 classes. 6 of the classes are white blood cells, while the remaining classes are erythroblasts and platelets. In this experiment, the dataset was sharded into one set containing the white blood cells and the other containing the erythroblasts and platelets. VGGs were trained on each set separately to distinguish the cell types within each set (i.e. one model to distinguish white blood cell types, and the other to distinguish between erythroblasts and platelets). The results of fusing these models are shown in Table 3. We can see that our model achieves meaningful transfer knowledge without requiring the sharing of data private data.

## 5.3 NON-IID SETUP

Similar to the "sharded" setup, the data is split disjointly between models. In this case, however, multiple models may receive samples from the same class, but with skewed class distributions. We again evaluate all algorithms in *zero-shot* fusion. Experiments are conducted on VGG11 with CIFAR-10, with results averaged over five random seeds. Since VGG models are no longer state-of-the-art, the main results have been moved to Appendix G.1.

## 5.4 FULL DATASET SETUP

Previous work on model fusion has predominantly considered the case where models are trained on the full dataset, followed by a fine-tuning phase aimed at achieving performance gains over the individual base models. We refer to this setting as the *full-dataset* setup, and we include it in

Table 3: **Test accuracy** comparison for VGGs fused on sharded splits of BloodMNIST. For each algorithm, we show the result with the importance scores that result in the best accuracy.

| Method | 2-WAY SPLIT |
|---|---|
| Individual Models | $76.0_{\pm 0.2}$ $22.8_{\pm 0.0}$ |
| Ensemble | $86.7_{\pm 4.8}$ |
| Vanilla Averaging | $13.3_{\pm 7.2}$ |
| KD | $54.8_{\pm 10.4}$ |
| LP | $75.6_{\pm 11.6}$ |
| OTFusion | $16.7_{\pm 5.3}$ |
| Git Re-Basin | $46.2_{\pm 9.6}$ |
| HF Linear (Ours) | $\mathbf{84.2}_{\pm 6.1}$ |

our evaluation for completeness. For Transformer architectures, to the best of our knowledge, the only existing baseline is Transformer OTFusion (Imfeld et al., 2023), which our approach consistently

Table 4: **Test accuracy** comparison for ViTs trained and fine-tuned on CIFAR-100 in the **Full-Dataset** setup. For the table with full details, please refer to Table 16.

| | | Zero-shot | | | Fine-tuning | |
| --- | --- | --- | --- | --- | --- | --- |
| | Base Models | Transformer OTFusion | K-means Gradient Fusion (Ours) | Ensemble | Transformer OTFusion | K-means Gradient Fusion (Ours) |
| 2-way fusion: | 73.9, 73.4 | $4.3_{\pm0.2}$ -69.6 | $63.0_{\pm1.2}$ -10.9 | $75.7_{\pm0.3}$ +1.8 | $74.0_{\pm0.4}$ +0.1 | $75.4_{\pm0.1}$ +1.5 |
| Inference Cost: | ×1 | ×1 | ×1 | ×2 | ×1 | ×1 |
| 4-way fusion: | 74.1, 73.6, 73.0, 72.9 | 1.0 -73.1 | 57.5 -16.6 | 76.6 +2.5 | 72.6 -1.5 | 75.6 +1.5 |
| Inference Cost: | ×1 | ×1 | ×1 | ×4 | ×1 | ×1 |

Table 5: **Test accuracy** comparison for ViTs trained and fine-tuned on Tiny-ImageNet in the **Full-Dataset** setup. For the table with full details, please refer to Table 18.

| | | Zero-shot | | | Fine-tuning | |
| --- | --- | --- | --- | --- | --- | --- |
| | Base Models | Transformer OTFusion | K-means Gradient Fusion (Ours) | Ensemble | Transformer OTFusion | K-means Gradient Fusion (Ours) |
| | 52.7, 51.7 | $3.1_{\pm0.2}$ -49.6 | $42.9_{\pm0.3}$ -9.8 | $54.9_{\pm0.4}$ +2.2 | $53.8_{\pm0.1}$ +1.1 | $54.2_{\pm0.4}$ +1.5 |
| Inference Cost: | ×1 | ×1 | ×1 | ×2 | ×1 | ×1 |

outperforms (see Table 4 and Table 5). Further details regarding both the fine-tuning and pre-training procedures are provided in Appendix F.3.

## 5.5 ROBUSTNESS STUDIES

In this subsection, we present several experiments designed to probe the performance of our algorithms under various scenarios.

### 5.5.1 VARYING FUSION DATASET SIZE

Gradient-based fusion algorithms typically require a substantial amount of data, which makes them sensitive to the size of the available fusion set. This limitation can be mitigated through the use of data augmentation. As shown in our ablation study on Table 6, augmenting a smaller fusion dataset (1k samples) substantially narrows the performance gap relative to using a larger dataset (5k samples).

Table 6: Zero-shot accuracy for two Non-IID VGG-11, when the fusion dataset size for KF Gradient is varied.

| Fusion Dataset Size | Model 1 | Model 2 | Uniform | Conductance | DeepLIFT |
| --- | --- | --- | --- | --- | --- |
| 5K samples | | | $76.1_{\pm0.5}$ | $76.2_{\pm0.2}$ | $76.3_{\pm0.4}$ |
| 1K samples | $73.2_{\pm1.2}$ | $71.3_{\pm1.1}$ | $73.2_{\pm0.5}$ | $71.7_{\pm0.8}$ | $71.4_{\pm0.7}$ |
| 1K samples + Augmentations | | | $74.5_{\pm0.8}$ | $75.3_{\pm0.6}$ | $75.2_{\pm0.5}$ |

### 5.5.2 RESNET COMPRESSION

Our fusion algorithm can be thought of as a compression algorithm, by *fusing a model into a smaller version of itself*. In this experiment, we train a ResNet34 on the full CIFAR-100 dataset and compress it into a ResNet18, using only 1/3 of the CIFAR-100 classes. The ResNet18 was initially trained on 1/3 of the classes, and the fusion was performed with KF using 5k samples from the same set of classes. Here, we only use the activations of the ResNet34 to form the target neurons. This can be seen as a form of stage-wise distillation of the larger ResNet into the smaller version. We compare our algorithm with Knowledge Distillation using the same fusion dataset in Table 7. Strikingly, our methods achieve an accuracy around 70% while only using 1/3 of the classes.

### 5.5.3 FUSED MODEL ANALYSIS AND INSIGHTS

In Fig. 2, we visualize the loss and accuracy landscapes of ViT base models trained on CIFAR-10, along with one of our fused models, using linear interpolation between model weights. The contour

Table 7: Zero-shot accuracy for ResNet18, obtained by compressing a ResNet34 with KF Gradient.

| ResNet18 | ResNet34 | Uniform | Conductance | DeepLIFT | Knowledge Distillation |
|----------|----------|---------|-------------|----------|------------------------|
| 29.49 | 82.35 | 70.04 | **70.44** | 69.82 | 31.46 |

plot reveals flatter basins around the fused model, which, as noted by Hochreiter and Schmidhuber (1997), is often indicative of improved generalization.

## 6 LIMITATIONS

Despite the strong performance of our proposed algorithms – often surpassing baselines and approaching that of ensembles – there remain areas for improvement.

First, the gradient-based variant of our approach is sensitive to hyperparameters and requires non-trivial tuning. While we were experimentally able to verify a set of hyperparameters that generalize well across our setups, this is not sufficient to claim universality.

Second, the effectiveness of our gradient-based fusion algorithm appears to scale with the size of the fusion dataset. While this dependency is encouraging in that more data yields better performance, it also highlights a shortcoming in the amount of data required. However using techniques such as data augmentation can close this gap, as shown in Table 6. Furthermore, recent work (Nasery et al., 2025) has explored doing fusion using open-source datasets, which could be an interesting direction for future work, to make our algorithms data-free.

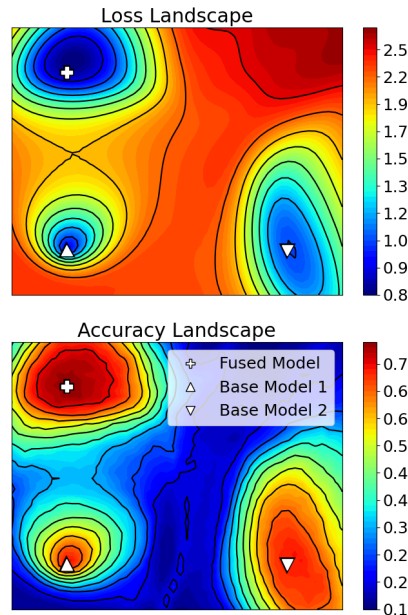

Figure 2: ViT Landscapes for CIFAR10, showing fused model from K-means Gradient Fusion using DeepLIFT scores

## 7 FUTURE WORK

While our approach demonstrates strong empirical performance across a variety of fusion scenarios, several avenues remain open for further exploration and refinement.

**Experiments with LLMs**. With the rise of LLMs, it would be interesting to see if our algorithms can yield improvements on such large-scale models, both in the context of fusion and compression.

**Automating fusion hyperparameter selection and level partitioning**. Future work could explore principled methods for automatically tuning fusion hyperparameters, including the choice of level granularity and whether to end levels at before or after activation functions.

**Other grouping methods**. Besides k-means clustering and matching, other methods to extract a layer-wise target could be explored. A simple idea would be to use the activations of neurons with the highest importance scores.

## 8 CONCLUSION

In this work, we introduced a novel neuron-aware approach to model fusion that supports fusing generic model architectures. Our algorithms, to our knowledge, are the first to successfully incorporate neuron importance scores in model fusion. Furthermore, our empirical results across diverse setups-including non-IID, sharded, and full-dataset regimes-consistently show that our fusion algorithms are competitive with or outperform existing baselines, especially in the zero-shot scenario, and in some cases approach ensemble-level performance.

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

## A  PROOFS

We first present the proof for Theorem 1.

*Proof.* We proceed to decompose the cost of Eq. (1) as follows, omitting the input $\mathbf{x}$ for notational clarity:

$$
\begin{aligned}
J_{\mathbf{w}} &= \sum_{j=1}^{d^{\mathcal{M}}} s_j \min_k \left\{ \left( z_k^{\mathcal{F}} - z_j \right)^2 \right\} \\
&= \sum_{j=1}^{d^{\mathcal{M}}} s_j \left( z_{k_j}^{\mathcal{F}} - z_j \right)^2 \quad \text{where } k_j = \arg\min_k \left( z_k^{\mathcal{F}} - z_j \right)^2 \implies j \in R_{k_j} \\
&= \sum_{j=1}^{d^{\mathcal{M}}} s_j \left[ \left( z_{k_j}^{\mathcal{F}} - T_{k_j} \right)^2 + 2 \left( z_{k_j}^{\mathcal{F}} - T_{k_j} \right) \left( T_{k_j} - z_j \right) + \left( T_{k_j} - z_j \right)^2 \right] \quad \left( \pm T_{k_j} \right) \quad (4)
\end{aligned}
$$

Here, when we set $k_j = \arg\min_k \left( z_k^{\mathcal{F}} - z_j \right)^2$, we break ties arbitrarily such that the sets $(R_k)_k^{d^{\mathcal{F}}}$ are non-overlapping and cover all base model neurons.

Since no constraints are imposed on the target vector $\mathbf{T}$, we retain the flexibility to define it in a manner that simplifies the optimization. Specifically, if we rearrange the summation in Eq. (4) into two nested summations – first over neurons in the fused model (i.e., $k = 1, \ldots, d^{\mathcal{F},i}$), and then over original neurons $j$ assigned to each $k$ (i.e., $k_j = \arg\min_k \left( z_k^{\mathcal{F}} - z_j \right)^2$) and define $T_k$ as the importance-weighted mean of the assigned level outputs, i.e., $T_k = \frac{\sum_{j \in R_k} s_j z_j}{\sum_{j \in R_k} s_j}$, then the cross-term

in Eq. (4) vanishes:

$$
\begin{aligned}
\sum_{j=1}^{d^{\mathcal{M}}} 2 s_j \left( z_{k_j}^{\mathcal{F}} - T_{k_j} \right) \left( T_{k_j} - z_j \right) &= 2 \sum_{k=1}^{d^{\mathcal{M}}} \sum_{j \in R_k} s_j \left( z_k^{\mathcal{F}} - T_k \right) \left( T_k - z_j \right) \\
&= 2 \sum_{k=1}^{d^{\mathcal{M}}} \left( z_k^{\mathcal{F}} - T_k \right) \sum_{j \in R_k} s_j \left( T_k - z_j \right) \\
&= 2 \sum_{k=1}^{d^{\mathcal{M}}} \left( z_k^{\mathcal{F}} - T_k \right) \sum_{j \in R_k} s_j \left( \frac{\sum_{i \in R_k} s_i z_i}{\sum_{i \in R_k} s_i} - z_j \right) \\
&= 2 \sum_{k=1}^{d^{\mathcal{M}}} \left( z_k^{\mathcal{F}} - T_k \right) \left( \sum_{i \in R_k} s_i z_i - \sum_{j \in R_k} s_j z_j \right) \\
&= 0
\end{aligned}
$$

Therefore, Eq. (4) becomes:

$$
\begin{aligned}
J_{\mathbf{w}} &= \sum_{j=1}^{d^{\mathcal{M}}} s_j \left[ \left( z_{k_j}^{\mathcal{F}} - T_{k_j} \right)^2 + \left( T_{k_j} - z_j \right)^2 \right] \\
&= \sum_{k=1}^{d^{\mathcal{M}}} \sum_{j \in R_k} s_j \left[ \left( z_k^{\mathcal{F}} - T_k \right)^2 + s_j \left( T_k - z_j \right)^2 \right] \qquad \text{(re-express sum over output neurons)}
\end{aligned}
$$

$\square$

We now present the proof for Theorem 2.

*Proof.* We analyze Hungarian Fusion and K-means Fusion separately.

**(a) Optimality of Hungarian Fusion:** As established in Section 4.1, the decoupled objective Eq. (2) separates into two terms: the *grouping error* and the *approximation error*. For linear levels, the layer outputs $z_k$ are affine functions of the weights $\mathbf{w}$, and thus the approximation error reduces to a weighted least squares problem, which admits a closed-form solution.

Consequently, minimizing the total cost reduces to minimizing the grouping error. In the special case of two models with equal-sized layers and one-to-one neuron matching, this corresponds to a Linear Sum Assignment Problem (LSAP) with importance-weighted squared error as the cost matrix. The Hungarian algorithm solves this problem exactly in polynomial time (Kuhn, 1955), hence the HF algorithm returns the optimal solution.

**(b) Approximation Bound for K-means Fusion:** We consider a fixed assignment of neurons, where we assign the $j^{th}$ base model neuron to the fused neuron $k_j$. Consider the total representation cost associated with all the base model neurons assigned to the $k^{th}$ fused neuron for the layer $l$. That is, the total representation cost of all base neurons $j$ that have $k_j = k \implies j \in R_k$. For a single sample, this is $\sum_{j \in R_k} s_j (z_k^{\mathcal{F}} - z_j)^2$. If we stack this over the samples, we get $\sum_{j \in R_k} s_j \|\mathbf{z}_k^{\mathcal{F}} - \mathbf{z}_j\|^2$, where $\mathbf{z}_k^{\mathcal{F}}, \mathbf{z}_k \in \mathbb{R}^n$ are column vectors with each entry corresponding to the preactivation for one input sample. We let the previous layer's activations be $\mathbf{X} \in \mathbb{R}^{n \times d^{\mathcal{M}}}$. Now, since it is a linear function of the previous layer's activations, we have $\mathbf{z}_k^{\mathcal{F}} = \mathbf{X} \mathbf{w}_k$, with $\mathbf{w}_k$ being the weights associated with the $k^{th}$ fused neuron (here we append a column of 1s to $X$ if we also have a bias term). Consider the projection matrix $\mathbf{P} = \mathbf{X}(\mathbf{X}^T \mathbf{X})^+ \mathbf{X}^T$ that projects vectors to the column space of $\mathbf{X}$. Then $\mathbf{I} - \mathbf{P}$ projects to the orthogonal complement of the image of $\mathbf{X}$. Recall that $\mathbf{P}\mathbf{X} = \mathbf{X}$ and $(\mathbf{I} - \mathbf{P})\mathbf{X} = 0$. We then have

$$\sum_{j \in R_k} s_j ||\mathbf{z}_k^{\mathcal{F}} - \mathbf{z}_j||^2 = \sum_{j \in R_k} s_j ||\mathbf{X}\mathbf{w}_k - \mathbf{z}_j||^2$$

$$= \sum_{j \in R_k} s_j ||\mathbf{P}(\mathbf{X}\mathbf{w}_k - \mathbf{z}_j) + (\mathbf{I} - \mathbf{P})(\mathbf{X}\mathbf{w}_k - \mathbf{z}_j)||^2$$

$$= \sum_{j \in R_k} s_j(||\mathbf{P}(\mathbf{X}\mathbf{w}_k - \mathbf{z}_j)||^2 + ||(\mathbf{I} - \mathbf{P})(\mathbf{X}\mathbf{w}_k - \mathbf{z}_j)||^2)$$

(due to orthogonality)

$$= \sum_{j \in R_k} s_j ||\mathbf{X}\mathbf{w}_k - \mathbf{P}\mathbf{z}_j||^2 + \sum_{j \in R_k} s_j ||(\mathbf{I} - \mathbf{P})\mathbf{z}_j||^2$$

We consider the term $\sum_{j \in R_k} s_j ||\mathbf{X}\mathbf{w}_k - \mathbf{P}\mathbf{z}_j||^2$. Letting $\bar{\mathbf{z}}_k = \frac{\sum_{j \in R_k} s_j \mathbf{z}_j}{\sum_{j \in R_k} s_j}$ (such that $\sum_{j \in R_k} s_j(\mathbf{P}\bar{\mathbf{z}}_k - \mathbf{P}\mathbf{z}_j) = 0$), we have

$$\sum_{j \in R_k} s_j ||\mathbf{X}\mathbf{w}_k - \mathbf{P}\mathbf{z}_j||^2 = \sum_{j \in R_k} s_j ||(\mathbf{X}\mathbf{w}_k - \mathbf{P}\bar{\mathbf{z}}_k) + (\mathbf{P}\bar{\mathbf{z}}_k - \mathbf{P}\mathbf{z}_j)||^2$$

$$= \sum_{j \in R_k} s_j ||\mathbf{X}\mathbf{w}_k - \mathbf{P}\bar{\mathbf{z}}_k||^2 + 2 \sum_{j \in R_k} s_j(\mathbf{X}\mathbf{w}_k - \mathbf{P}\bar{\mathbf{z}}_k)^T(\mathbf{P}\bar{\mathbf{z}}_k - \mathbf{P}\mathbf{z}_j)$$

$$+ \sum_{j \in R_k} s_j ||\mathbf{P}\bar{\mathbf{z}}_k - \mathbf{P}\mathbf{z}_j||^2$$

$$= \sum_{j \in R_k} s_j ||\mathbf{X}\mathbf{w}_k - \mathbf{P}\bar{\mathbf{z}}_k||^2 + 2(\mathbf{X}\mathbf{w}_k - \mathbf{P}\bar{\mathbf{z}}_k)^T \underbrace{\sum_{j \in R_k} s_j(\mathbf{P}\bar{\mathbf{z}}_k - \mathbf{P}\mathbf{z}_j)}_{0}$$

$$+ \sum_{j \in R_k} s_j ||\mathbf{P}\bar{\mathbf{z}}_k - \mathbf{P}\mathbf{z}_j||^2$$

$$= \sum_{j \in R_k} s_j ||\mathbf{X}\mathbf{w}_k - \mathbf{P}\bar{\mathbf{z}}_k||^2 + \sum_{j \in R_k} s_j ||\mathbf{P}\bar{\mathbf{z}}_k - \mathbf{P}\mathbf{z}_j||^2$$

Thus, substituting this back, and summing over $k$ to get the whole layer's representation cost, we get

$$\sum_{k=1}^{d^{\mathcal{F}}} \sum_{j \in R_k} s_j ||\mathbf{z}_k^{\mathcal{F}} - \mathbf{z}_j||^2 = \sum_{k=1}^{d^{\mathcal{F}}} \sum_{j \in R_k} s_j ||\mathbf{X}\mathbf{w}_k - \mathbf{P}\bar{\mathbf{z}}_k||^2$$

$$+ \sum_{k=1}^{d^{\mathcal{F}}} \sum_{j \in R_k} s_j ||\mathbf{P}\bar{\mathbf{z}}_k - \mathbf{P}\mathbf{z}_j||^2 + \sum_{j=1}^{d^{\mathcal{M}}} s_j ||(\mathbf{I} - \mathbf{P})\mathbf{z}_j||^2$$

Note first that the last term in the sum on the right is always incurred independently of the assignment $k_j$ or the chosen weights $\mathbf{w}_k$.

Assume we have a solution that obtains the optimal representation cost $OPT$. Then, since the first term in the sum is nonnegative, the representative cost of the optimal solution is at least the optimal value of $\sum_{k=1}^{d^{\mathcal{F}}} \sum_{j \in R_k} s_j ||\mathbf{P}\bar{\mathbf{z}}_k - \mathbf{P}\mathbf{z}_j||^2 + \sum_{j=1}^{d^{\mathcal{M}}} s_j ||(\mathbf{I} - \mathbf{P})\mathbf{z}_j||^2$. If we let $OPT_{grouping}$ be the minimum possible value of $\sum_{k=1}^{d^{\mathcal{F}}} \sum_{j \in R_k} s_j ||\mathbf{P}\bar{\mathbf{z}}_k - \mathbf{P}\mathbf{z}_j||^2$, then we get $OPT \geq OPT_{grouping} + \sum_{j=1}^{d^{\mathcal{M}}} s_j ||(\mathbf{I} - \mathbf{P})\mathbf{z}_j||^2$

We now consider the sum on the right for KF. Recall that KF first projects the activations to the image of $X$ and then finds the K-means clusters. That is, it finds the K-means clustering of $(\mathbf{P}\mathbf{z}_j)_{j=1}^{d^{\mathcal{M}}}$

to find the clusters $R_k$, aiming to minimize $\sum_{k=1}^{d^{\mathcal{F}}} \sum_{j \in R_k} s_j ||\mathbf{P}\bar{\mathbf{z}}_k - \mathbf{P}\mathbf{z}_j||^2$. Then, it fits $\mathbf{w}_k$ such that $||\mathbf{X}\mathbf{w}_k - \mathbf{P}\bar{\mathbf{z}}_k||^2$ is minimized by solving a weighted least squares problem as elaborated in Appendix B.2. Notice that this achieves $||\mathbf{X}\mathbf{w}_k - \mathbf{P}\bar{\mathbf{z}}_k||^2 = 0$, since, as $\mathbf{P}\bar{\mathbf{z}}_k$ is in the image of $\mathbf{X}$ by virtue of being a projection to that image, there is a $\mathbf{w}_k$ satisfying $\mathbf{X}\mathbf{w}_k = \mathbf{P}\bar{\mathbf{z}}_k$. Thus, the total representation cost for KF for any layer will be $\sum_{k=1}^{d^{\mathcal{F}}} \sum_{j \in R_k} s_j ||\mathbf{P}\bar{\mathbf{z}}_k - \mathbf{P}\mathbf{z}_j||^2 + \sum_{j=1}^{d^{\mathcal{M}}} s_j ||(\mathbf{I} - \mathbf{P})\mathbf{z}_j||^2$, where the first term is the the weighted K-means cost with respect to clustering the projected preactivations, and the second term is always incurred regardless of assignment or chosen weights.

The K-means problem is NP-hard in general (Aloise et al., 2009). However, the local-search algorithm introduced by Kanungo et al. (2002), provides a $(9 + \epsilon)$-approximation guarantee for the weighted K-means cost under squared Euclidean distance. By using this algorithm to construct the cluster assignments in KF, we obtain a solution where the term $\sum_{k=1}^{d^{\mathcal{F}}} \sum_{j \in R_k} s_j ||\mathbf{P}\bar{\mathbf{z}}_k - \mathbf{P}\mathbf{z}_j||^2$ is within a constant factor of the optimal cost $OPT_{grouping}$.

Thus, the cost for this layer attained by KF is at most $(9+\epsilon)OPT_{grouping} + \sum_{j=1}^{d^{\mathcal{M}}} s_j ||(\mathbf{I}-\mathbf{P})\mathbf{z}_j||^2 \leq (9 + \epsilon)(OPT_{grouping} + \sum_{j=1}^{d^{\mathcal{M}}} s_j ||(\mathbf{I} - \mathbf{P})\mathbf{z}_j||^2) \leq (9 + \epsilon)OPT$, showing that it is a $(9 + \epsilon)$-approximation for the layer representation cost.

$\square$

# B   EFFICIENTLY MINIMIZING THE FUSION ERRORS

## B.1   MINIMIZING THE GROUPING ERROR

For the special case (a), we can re-express Eq. (1) as a sum over the two base models:

$$J_{\mathbf{w}} = \sum_{j=1}^{d^{M_1}} s_j^{M_1} \min_k \left\{ \left( z_k^{\mathcal{F}} - z_j^{M_1} \right)^2 \right\} + \sum_{j=1}^{d^{M_2}} s_j^{M_2} \min_k \left\{ \left( z_k^{\mathcal{F}} - z_j^{M_2} \right)^2 \right\} \tag{5}$$

This cost can also be decomposed analogously to Eq. (2). We can now define the cost of matching neuron $j_1$ of $M_1$ to neuron $j_2$ of $M_2$, as the cost of trying to approximate the resulting cluster center $T_{j_1, j_2}$, from any neuron of the fused model $\mathcal{F}$. A simpler alternative/heuristic is to just compute the distances between the level outputs. After defining this cost matrix, we can then run the Hungarian Matching algorithm (Kuhn, 1955) to find a one-to-one matching that minimizes Eq. (5).

For the general case (b), we can run Lloyd's (Lloyd, 1982) algorithm for K-means, since it usually offers a good tradeoff between simplicity and effectiveness. By making use of the K-means++ initialization, we usually get better clusterings. Note that for this task, we treat neurons "as data points", in the sense that we want to cluster neurons together. Therefore, the features of a neuron are the values (e.g. activations) it takes for different samples $\mathbf{x}$ in the dataset. Clustering is then performed over these vectors using importance-weighted K-means, where the number of clusters "k" is set to the desired number of neurons in the fused layer. Once clusters are formed, we compute the corresponding importance-weighted centroids, giving us the target matrix $\mathbf{T} \in \mathbb{R}^{B \times d^{\mathcal{F}}}$, where $B$ is the batch dimension.

## B.2   MINIMIZING THE APPROXIMATION ERROR

For the special case where a level is a linear function of its weights $\mathbf{w}$, i.e. $\mathbf{z} = \mathbf{X}\mathbf{w}$ for some $\mathbf{X}$, then the approximation error in Eq. (3) admits to a closed-form weighted-MSE solution:

$$\mathbf{w}^* = \left( \mathbf{X}^\top \mathbf{S} \mathbf{X} \right)^+ \mathbf{X}^\top \mathbf{S} \mathbf{T}$$

where $\mathbf{S} = \text{diag}(s_1, \ldots, s_{d^{\mathcal{M}}})$, and $()^+$ denotes the Moore-Penrose pseudoinverse.

For the general case, where a level is a non-linear differentiable function of its weights, we can obtain a local minima by optimizing with SGD. In practice we often use Adam (Kingma and Ba, 2014) or AdamW (Loshchilov and Hutter, 2017). Furthermore, in practice, we do not minimize the weighted MSE, but rather the plain MSE. This is due to the fact that neurons with low importance

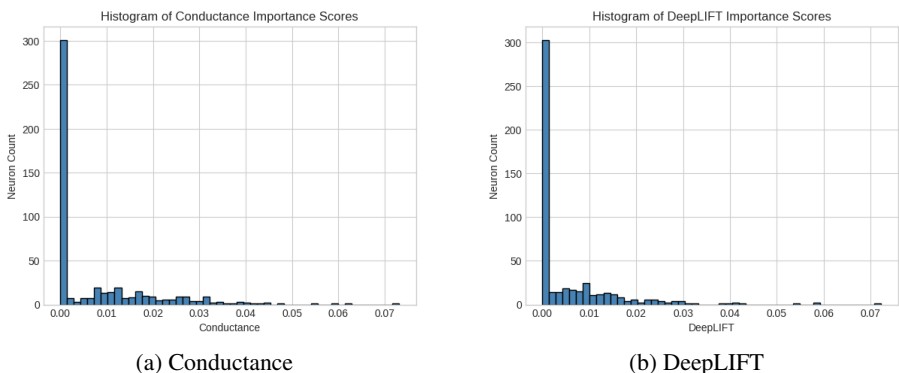

(a) Conductance         (b) DeepLIFT

Figure 3: Histogram of Conductance and DeepLIFT Importance Scores

scores will barely change if we use the weighted MSE. While this could only minimally affect the representation loss for the current level, it could lead to noisy inputs in later levels, or just poor intermediate representations. In our experiments we noticed that, especially in non-IID cases, many neurons tend to be attributed scores that are virtually zero as seen in Fig. 3.

## C    COMPARISON OF HUNGARIAN FUSION WITH EXISTING ALGORITHMS

We note that **Hungarian Fusion** is in spirit very similar to both **OTFusion** (Singh and Jaggi, 2020) and the activations-based version of **Git Re-Basin** (Ainsworth et al., 2022). In the case of **Equal-size models with level-wise one-to-one matching** that we restrict HF to, they all construct a matching between neurons (or equivalent a transport map or permutation matrix to align the second model to the first) by solving a minimum cost matching problem.

However, a key difference is that HF accounts for the effect of refitting the previous layers and correspondingly refits the weights of the current layer being considered to minimize the effect of the accumulated error. Empirically, this significantly improves the zero-shot performance significantly as shown in our experimental results.

With regards to the incorporation of neuron importance scores, for OTFusion, Singh and Jaggi (2020) proposed using the neuron importance as the probability measure assigned to a neuron in the optimal transport problem setup, while Ainsworth et al. (2022) did not discuss applying importance scores in Git Re-Basin. In our experiments, we follow the recommendation of Singh and Jaggi (2020) for OTFusion and for Git Re-Basin we weigh the neuron weights according to the neuron's score when averaging, in the same manner as we do for HF.

## D    DATA PARTITIONING REGIMES

### D.1    NON-IID SPLITS

To simulate non-IID splits, we utilize a Dirichlet distribution to create unbalanced class distributions across models. Let $N_c$ represent the number of data points in class $c$, and $\alpha_k$ denote the concentration parameter for model $k$. The data for each class $c$ is distributed across models as:

$$\text{split}_k \sim \text{Dir}(\alpha_1, \ldots, \alpha_k)$$

where $\text{Dir}(\cdot)$ represents the Dirichlet distribution. The concentration parameters $\alpha_k$ are arranged in an ordered sequence, determined by the parameter min_max_ratio:

```
alpha_min = 1.0
min_max_ratio = 0.2
alpha_max = alpha_min / min_max_ratio
alphas = linspace(alpha_min, alpha_max, min_max_ratio)
```

A smaller ratio results in a wider disparity between splits, amplifying the heterogeneity. The Dirichlet-distributed probabilities dictate the number of samples assigned to each model for class $c$, ensuring that splits exhibit diverse and non-uniform class distributions. Random shuffling of *class indices* and *concentration parameters* for each class $c$ introduces additional randomness in the resulting splits.

## D.2 SHARDED SPLITS

In the sharded partitioning regime, the dataset is split such that each model receives examples from a disjoint subset of classes. That is, no two models share any classes in their local datasets, simulating a strongly divergently distributed base model training datasets scenario based on class exclusivity rather than distributional imbalance.

Let $\mathcal{C}$ denote the set of all classes in the dataset. The class set is first randomly permuted and then evenly partitioned into $K$ disjoint subsets, where $K$ is the number of models. Each subset $\mathcal{C}_k$ is assigned to model $k$, and all examples belonging to classes in $\mathcal{C}_k$ are included in that model's local dataset:

$$\bigcup_{k=1}^{K} \mathcal{C}_k = \mathcal{C}, \quad \mathcal{C}_i \cap \mathcal{C}_j = \emptyset \quad \forall i \neq j$$

## E  MODEL TRAINING DETAILS

We trained **VGGs on CIFAR-10** and **ViTs on CIFAR100**. All models were trained on NVIDIA RTX A5000 GPUs.

The VGGs followed the VGG11 architecture and the implementation is based on the open source implementation provided by [1]Singh and Jaggi (2020).

The ViTs implementation is based on [2]omihub777, and we used the following model hyperparameters:

| Model Hyperparameter | Value |
| --- | --- |
| Patch Size | 8 |
| Attention Heads | 12 |
| Encoder Blocks | 7 |
| Feed Forward Network Hidden Size | 384 |
| Encoder Hidden Size | 384 |

To train the models, we use the non-exhaustive list of hyperparameters listed in Table 8

| Splits | VGG Epochs | ViT Epochs | | Training Hyperparameter | Value |
| --- | --- | --- | --- | --- | --- |
| Full Dataset | 300 | 350 | | Warmup Epochs | 5 |
| Split By 2 | 200 | 250 | | Minimum Learning Rate | $10^{-5}$ |
| Split By 4 | 150 | 225 | | Learning Rate | $10^{-3}$ |
| Split By 6 | – | 200 | | Label Smoothing | 0.1 |
| Split By 8 | 125 | – | | Batch Size | 128 |

(a) Number of training epochs  (b) Training hyperparameters

Table 8: Training configurations and schedules for VGG and ViT models.

The following torch augmentations were used for training: RandomCrop, RandomHorizonalFlip, Normalize, and other augmentations as in omihub777 based on Cubuk et al. (2018).

The model was trained using torch's Gradual Warmup Scheduler with torch's CosineAnnealingLR scheduler.

---

[1]https://github.com/sidak/otfusion
[2]https://github.com/omihub777/ViT-CIFAR

We believe that these are sufficient to reproduce the main claims of our work. Additional information about hyperparameters can be found in our open-source code repository.

## F  FUSION IMPLEMENTATION DETAILS

### F.1  FUSION HYPERPARAMETERS

Our proposed fusion algorithms include both linear and gradient-based variants, each with distinct hyperparameter considerations.

**Linear Variants.** The linear fusion algorithms (e.g., plain Hungarian Fusion and K-means Fusion) require minimal hyperparameter tuning. The primary decisions involve whether to normalize (i) neuron outputs and/or (ii) neuron importance scores. In our experiments, we found that omitting normalization typically yielded better results across both all data partitioning settings. This is likely because normalization can distort relative differences in neuron output magnitude that are informative for matching or clustering.

**Gradient-based Variants.** In contrast, the gradient-based fusion variant introduces a broader set of hyperparameters. These include:

1. **Optimization Parameters:** learning rate, weight decay, number of gradient steps per level, and batch size, validation split, validation patience.
2. **Initialization Scheme:** initialization of the fused model weights at each level (e.g., weights from a randomly selected base model with added noise $\epsilon$).
3. **Clustering Settings:** number of clusters (typically matched to the fused model's layer width), use of K-means++ initialization, early stopping criteria and whether to normalize neuron outputs just for the clustering stage.
4. **Importance Weighting:** whether and how to incorporate neuron importance scores into both clustering and loss weighting.

While this added complexity increases flexibility and modeling capacity, it also requires careful tuning for stable and effective optimization. To mitigate this, we conducted extensive experiments and identified two sets of hyperparameter configurations that generalized well across datasets (CIFAR-10, CIFAR-100), model architectures (VGG11, ViT), and fusion regimes (Full Dataset, Non-IID, Sharded). Specifically, we found the hyperparameters in Table 9.

Table 9: Sets of hyperparameters used for K-means Gradient Fusion

| Hyperparameter | Setting 1 | Setting 2 |
|---|---|---|
| Optimizer ($n-1$ first levels) | AdamW | SGD |
| Learning Rate ($n-1$ first levels) | $10^{-3}$ | $10^{-4}$ |
| Epochs ($n-1$ first levels) | 100 | 50 |
| Weight Decay (All levels) | $10^{-4}$ | $10^{-4}$ |
| Perturbation $\epsilon$ | 1.0 | 0.1 |
| Optimizer (Last level) | AdamW | Adam |
| Learning Rate (Last level) | $10^{-3}$ | $10^{-3}$ |
| Epochs (Last level) | 100 | 100 |
| Epochs ($n-1$ first levels) | 100 | 100 |
| Normalize Activations | False | False |
| Train Batch Size | 32 | 32 |
| Val Split | 0.1 | 0.1 |
| Head Weights | False | False |

We note that "Head Weights" refers to weighing the final logits of the models by the proportion of samples seen per class, for every model. In practice, this heuristic improves accuracy by a small margin, but this comes at a cost of calibration, as the test loss increases, which in some cases might not be a good tradeoff.

Each setting of hyperparameters induces a different behavior in the gradient-based variant of our algorithm. By using Setting 1, we essentially take larger gradient steps that move us far away from initialization. The resulting model is quite different from base models, in terms of plain weight L2 norm. On the other hand, Setting 2 relies on the initialization of the model to be already decent (e.g. any base model), and takes small gradient steps. Empirically, we found that with the second setting, the majority of performance gain occurs at the classification head, where the targets become the raw average logits of all base models. This is similar to Linear Probing (LP), with the only difference that LP typically minimizes some sort of KL-divergence loss between softmaxed logits and average-softmaxed ensemble-logits, instead of minimizing the L2 distance between averaged raw logits. Nevertheless, most interesting models are produced with the first setting, which finds new solutions far away from initialization, and within a much richer context.

For the gradient variants of our algorithms, in our experiments:

- **All** Full-dataset models were fused using *Setting 1*.

- **All** sharded models were fused using *Setting 1*.

- **All** Non-IID models, **except** for VGG11s with $n = 2$ models for CIFAR-10 (which used Setting 1), were fused using *Setting 2*.

### F.2 MODEL PARTITIONING SCHEMES

Due to the flexibility of our algorithm, we had the freedom to develop our own partition. In practice we as we primarily tested on like models, there were obvious answers that we used.

For VGG11s, each level contained only a single convolutional or linear layer to be aligned or an activation function, which did not need to be aligned.

For ViTs, each level corresponded to an encoder block except for the last one which corresponded to the classifier head.

### F.3 POST FUSION FINETUNING HYPERPARAMETERS

For fusing full dataset models, a finetuning phase is shown to improve fused model performance above base model performance. For this finetuning phase we used the same optimizer that was used to train the corresponding base models, and torch's CosineAnnealingWarmRestarts scheduler. The whole process had the following hyperparameters:

| Hyperparameter | CIFAR 10/100 | Tiny-ImageNet |
|---|---|---|
| Learning Rate | $3 \cdot 10^{-4}$ | $10^{-5}$ |
| Minimum Learning Rate | $10^{-6}$ | $10^{-6}$ |
| Label Smoothing | 0.1 | 0.1 |
| Epochs | 200 | 200 |

The augmentation was the same generic suite as we used to train VGGs and ViTs initially. See Appendix E for more details. Once again, all code for reproduction is present in our open source repository.

### F.4 NEURON IMPORTANCE SCORE DETAILS

#### F.4.1 IMPLEMENTATION

For the computation of neuron importance scores, we used the LayerConductance (Dhamdhere et al., 2018) and LayerDeepLIFT (Shrikumar et al., 2017) implementations of Kokhlikyan et al. (2020). However, our fusion framework allows for the usage of any importance score, possibly computed by other means.

### F.4.2 Computation

Neuron importance scores can be estimated either (i) independently by each model using its own training or validation data, or (ii) jointly using the designated fusion dataset prior to fusion. The first approach typically yields more reliable estimates and, in our experiments, resulted in higher zero-shot accuracy. Moreover, it aligns naturally with the federated learning setting, where clients could compute scores locally and transmit them together with their models for fusion.

For our benchmarks, however, we adopt the second approach, as it is more computationally efficient: the fusion dataset is usually much smaller than the private datasets of the individual models.

### F.4.3 Score Selection

The choice of importance score can be regarded as a hyperparameter, akin to learning rate or weight decay, and can in principle be optimized using standard procedures such as cross-validation. In our experiments, uniform scores rarely outperformed other measures such as Conductance or DeepLIFT. The latter two usually performed on par, with Conductance showing a slight advantage in some cases.

### F.5 Algorithm Runtime Comparison

As our methods are technically specific implementations of our general framework, it is difficult to definitively give an evaluation of the overall framework. However, we did quantify the performance of our realizations, both on VGGs and ViTs. We only used importance scores for VGGs as they just add a constant startup time and usually do not interfere with the performance of the algorithm. For the same reason, we only use conductance when when testing for importance score times. Interestingly, KF Linear speeds up significantly when we use importance scores, suggesting that the K-means portion of the algorithm resolves faster in this case because of the weights. All experiments were ran on NVIDIA RTX A5000 GPUs.

Table 10: **Algorithm runtime** comparison when fusing VGG networks on CIFAR-10. We fused the same two models 10 times and averaged the run times. All algorithms were run with the same 400 samples in each iteration. All times are in seconds.

| Algorithm | Runtime (Uniform) | Runtime (Conductance) |
|---|---|---|
| OT Fusion | 0.7 | 3.0 |
| Git Re-Basin | 1.0 | 3.2 |
| HF Linear (Ours) | 14.2 | 16.5 |
| KF Linear (Ours) | 78.3 | 83.7 |
| KF Gradient Uniform (Ours) | 16.5 | 18.8 |

Table 11: **Algorithm runtime** comparison when fusing ViT networks on CIFAR-100. We fused the same two models 5 times using our KF Gradient method with uniform importance scores and averaged the run times.

| Fusion Samples | Runtime (s) |
|---|---|
| 400 | 38.1 |
| 6000 | 632.5 |

# G    ADDITIONAL RESULTS

In this section, besides complimentary tables, we will also present the full tables for results shown earlier. These full tables include the standard deviation for base models, as well as the performance of each fusion algorithm for each neuron importance score.

## G.1    VGGS ON CIFAR-10

Table 12: **Test accuracy** comparison when fusing VGG11 networks on CIFAR-10 for **Non-IID** splits. Fusion was performed using 400 data points sampled from the dataset seen by the first model. The same fusion data was used for all algorithms.

| Method | 2-WAY SPLIT | 4-WAY SPLIT | 8-WAY SPLIT |
|---|---|---|---|
| Individual Models | $83.8_{\pm2.7}$, $77.3_{\pm2.1}$ | $79.8_{\pm3.2}$, $77.5_{\pm2.9}$, $74.7_{\pm3.3}$, $69.7_{\pm4.6}$ | $72.3_{\pm1.8}$, $70.6_{\pm2.4}$, $67.7_{\pm1.1}$, $66.1_{\pm1.9}$ $65.4_{\pm1.6}$, $63.4_{\pm1.2}$, $58.6_{\pm3.5}$, $55.6_{\pm3.0}$ |
| Ensemble | $89.1_{\pm0.4}$ | $85.7_{\pm0.2}$ | $78.9_{\pm0.8}$ |
| Vanilla Averaging | $11.5_{\pm1.5}$ | $10.0_{\pm0.0}$ | $10.0_{\pm0.0}$ |
| KD | $83.3_{\pm1.5}$ | $79.2_{\pm1.8}$ | $71.4_{\pm1.4}$ |
| LP | $85.8_{\pm1.7}$ | $\mathbf{81.7}_{\pm1.7}$ | $\mathbf{74.2}_{\pm1.0}$ |
| OTF Uniform | $50.0_{\pm6.2}$ | $14.7_{\pm5.5}$ | $12.3_{\pm3.1}$ |
| OTF Conductance | $40.6_{\pm2.7}$ | $11.3_{\pm2.5}$ | $10.9_{\pm1.0}$ |
| OTF DeepLIFT | $41.2_{\pm3.1}$ | $12.7_{\pm2.8}$ | $14.0_{\pm2.0}$ |
| Git Re-Basin[1] Uniform | $58.0_{\pm3.3}$ | N/A | N/A |
| Git Re-Basin Conductance | $73.1_{\pm3.2}$ | N/A | N/A |
| Git Re-Basin DeepLIFT | $75.8_{\pm2.9}$ | N/A | N/A |
| HF Linear Uniform (Ours) | $78.0_{\pm2.6}$ | N/A | N/A |
| HF Linear Conductance (Ours) | $\mathbf{86.6}_{\pm0.5}$ | N/A | N/A |
| HF Linear DeepLIFT (Ours) | $86.5_{\pm0.5}$ | N/A | N/A |
| KF Linear Uniform (Ours) | $85.3_{\pm1.2}$ | $78.7_{\pm0.5}$ | $69.1_{\pm1.5}$ |
| KF Linear Conductance (Ours) | $86.5_{\pm0.6}$ | $79.5_{\pm0.8}$ | $71.3_{\pm1.4}$ |
| KF Linear DeepLIFT (Ours) | $86.5_{\pm0.3}$ | $79.6_{\pm0.8}$ | $71.3_{\pm1.2}$ |
| HF Gradient Uniform (Ours) | $85.4_{\pm1.9}$ | N/A | N/A |
| HF Gradient Conductance (Ours) | $85.5_{\pm2.0}$ | N/A | N/A |
| HF Gradient DeepLIFT (Ours) | $85.5_{\pm2.0}$ | N/A | N/A |
| KF Gradient Uniform (Ours) | $85.5_{\pm1.9}$ | $81.3_{\pm1.8}$ | $73.7_{\pm1.1}$ |
| KF Gradient Conductance (Ours) | $85.4_{\pm2.0}$ | $81.4_{\pm1.8}$ | $73.7_{\pm1.1}$ |
| KF Gradient DeepLIFT (Ours) | $85.4_{\pm2.0}$ | $81.3_{\pm2.0}$ | $73.8_{\pm1.1}$ |

---

[1]Git Re-Basin reduces to OTFusion when solving the OT problem exactly with uniform importance scores. In practice, OTFusion uses preactivations (Singh and Jaggi, 2020), while Git Re-Basin uses activations (Ainsworth et al., 2022); we follow these defaults. Empirically, both yield the same fused model when using preactivations.

Table 13: **Test accuracy** comparison when fusing VGG11 networks on CIFAR-10 for **Sharded** splits. Fusion was performed using 400 data points sampled from the dataset seen by the first model. The same fusion data was used for all algorithms.

| Method | 2-WAY SPLIT | 4-WAY SPLIT | 6-WAY SPLIT |
|---|---|---|---|
| Individual Models | $47.8_{\pm0.5}$, $46.7_{\pm0.8}$ | $29.1_{\pm0.1}$, $28.8_{\pm0.2}$, $19.7_{\pm0.2}$, $19.1_{\pm0.6}$ | $19.9_{\pm0.0}$, $19.8_{\pm0.1}$, $19.5_{\pm0.2}$, $15.2_{\pm3.7}$ $10.0_{\pm0.0}$, $10.0_{\pm0.0}$ |
| Ensemble | $80.2_{\pm2.3}$ | $58.3_{\pm1.7}$ | $41.3_{\pm1.7}$ |
| Vanilla Averaging | $12.1_{\pm2.2}$ | $10.0_{\pm0.0}$ | $10.0_{\pm0.0}$ |
| KD | $58.5_{\pm1.9}$ | $42.8_{\pm2.3}$ | $33.4_{\pm1.4}$ |
| LP | $49.8_{\pm1.2}$ | $32.0_{\pm1.7}$ | $22.0_{\pm1.0}$ |
| OTF Uniform | $28.3_{\pm5.9}$ | $11.5_{\pm2.1}$ | $10.4_{\pm0.8}$ |
| OTF Conductance | $24.9_{\pm6.3}$ | $10.8_{\pm1.4}$ | $10.0_{\pm0.0}$ |
| OTF DeepLIFT | $24.9_{\pm5.0}$ | $10.0_{\pm0.0}$ | $10.0_{\pm0.0}$ |
| Git Re-Basin Uniform | $30.6_{\pm4.2}$ | N/A | N/A |
| Git Re-Basin Conductance | $58.2_{\pm3.9}$ | N/A | N/A |
| Git Re-Basin DeepLIFT | $62.3_{\pm5.3}$ | N/A | N/A |
| HF Linear Uniform (Ours) | $60.5_{\pm2.9}$ | N/A | N/A |
| HF Linear Conductance (Ours) | $76.7_{\pm4.2}$ | N/A | N/A |
| HF Linear DeepLIFT (Ours) | $76.6_{\pm4.0}$ | N/A | N/A |
| KF Linear Uniform (Ours) | $\mathbf{77.1}_{\pm1.3}$ | $52.4_{\pm2.7}$ | $\mathbf{35.5}_{\pm3.2}$ |
| KF Linear Conductance (Ours) | $76.4_{\pm4.4}$ | $44.0_{\pm2.7}$ | $29.9_{\pm2.2}$ |
| KF Linear DeepLIFT (Ours) | $76.4_{\pm4.1}$ | $44.4_{\pm2.2}$ | $30.4_{\pm2.0}$ |
| HF Gradient Uniform (Ours) | $59.0_{\pm3.1}$ | N/A | N/A |
| HF Gradient Conductance (Ours) | $70.8_{\pm2.5}$ | N/A | N/A |
| HF Gradient DeepLIFT (Ours) | $70.3_{\pm1.5}$ | N/A | N/A |
| KF Gradient Uniform (Ours) | $69.7_{\pm1.9}$ | $45.9_{\pm2.6}$ | $34.7_{\pm3.4}$ |
| KF Gradient Conductance (Ours) | $71.3_{\pm1.1}$ | $44.1_{\pm2.7}$ | $34.7_{\pm1.8}$ |
| KF Gradient DeepLIFT (Ours) | $71.5_{\pm1.9}$ | $44.8_{\pm3.2}$ | $34.2_{\pm1.0}$ |

Table 14: **Test accuracy** comparison when fusing VGG11 networks pairwise on CIFAR-10 trained on the **full dataset**. Results are averaged across 3 seeds. Fusion was performed using 400 samples from the full dataset. The same fusion data was used for all algorithms. Fine-tuning was performed for 200 epochs with a learning rate of $3 \cdot 10^{-4}$ and a cosine annealing with warm restarts scheduler with a minimum learning rate of $10^{-6}$.

| Method | ZERO-SHOT | FINETUNED |
|---|---|---|
| Individual Models | $93.2_{\pm0.1}$ $93.0_{\pm0.1}$ | $93.2_{\pm0.1}$ $93.2_{\pm0.1}$ |
| Vanilla Averaging | $9.3_{\pm1.4}$ | − |
| Ensemble | $94.1_{\pm0.1}$ | $94.2_{\pm0.1}$ |
| OTF Uniform | $72.6_{\pm5.2}$ | $93.2_{\pm0.2}$ |
| OTF Conductance | $46.7_{\pm11.1}$ | $93.5_{\pm0.3}$ |
| OTF DeepLIFT | $49.8_{\pm4.0}$ | $93.4_{\pm0.1}$ |
| Git Re-Basin Uniform | $77.1_{\pm3.6}$ | $93.5_{\pm0.1}$ |
| Git Re-Basin Conductance | $61.8_{\pm2.8}$ | $93.3_{\pm0.1}$ |
| Git Re-Basin DeepLIFT | $65.3_{\pm5.9}$ | $\mathbf{93.6}_{\pm0.0}$ |
| HF Linear Uniform (Ours) | $87.0_{\pm0.2}$ | $93.3_{\pm0.1}$ |
| HF Linear Conductance (Ours) | $74.3_{\pm1.8}$ | $93.3_{\pm0.1}$ |
| HF Linear DeepLIFT (Ours) | $73.4_{\pm1.7}$ | $93.4_{\pm0.2}$ |
| KF Linear Uniform (Ours) | $74.2_{\pm0.8}$ | $93.2_{\pm0.2}$ |
| KF Linear Conductance (Ours) | $74.9_{\pm1.4}$ | $93.4_{\pm0.1}$ |
| KF Linear DeepLIFT (Ours) | $75.3_{\pm0.5}$ | $93.4_{\pm0.2}$ |
| HF Gradient Uniform (Ours) | $\mathbf{88.1}_{\pm0.2}$ | $93.4_{\pm0.1}$ |
| HF Gradient Conductance (Ours) | $87.9_{\pm0.2}$ | $93.4_{\pm0.1}$ |
| HF Gradient DeepLIFT (Ours) | $88.1_{\pm0.1}$ | $93.3_{\pm0.3}$ |
| KF Gradient Uniform (Ours) | $85.0_{\pm0.2}$ | $93.0_{\pm0.2}$ |
| KF Gradient Conductance (Ours) | $85.9_{\pm0.7}$ | $93.2_{\pm0.1}$ |
| KF Gradient DeepLIFT (Ours) | $86.2_{\pm1.3}$ | $93.0_{\pm0.1}$ |

## G.2 ViTs on CIFAR-100 and Tiny-ImageNet

### G.2.1 CIFAR-100

Table 15: **Test accuracy** comparison when fusing ViT networks on CIFAR-100 for **Sharded** splits. Fusion was performed using 5000 data points sampled from the dataset seen by the first model. For "activations-based" (i.e. acts) Transformer OTFusion, following (Imfeld et al., 2023), we used a subset of 200 samples. The weights-based variant (wts) does not use data. This table is complimentary to Table 2.

| Method | 2-way split | 4-way split | | 6-way split | | |
|---|---|---|---|---|---|---|
| Individual Models | $38.6_{\pm 0.5}$ $37.2_{\pm 0.6}$ | $20.4_{\pm 0.2}$ $19.5_{\pm 0.2}$ | $19.9_{\pm 0.1}$ $19.2_{\pm 0.3}$ | $14.3_{\pm 0.2}$, $13.2_{\pm 0.2}$ | $13.7_{\pm 0.3}$ $12.8_{\pm 0.3}$ | $13.5_{\pm 0.2}$ $12.2_{\pm 0.6}$ |
| Ensemble | $63.7_{\pm 0.4}$ | $53.4_{\pm 1.8}$ | | $45.4_{\pm 2.0}$ | | |
| Vanilla Averaging | $2.2_{\pm 0.6}$ | $1.4_{\pm 0.2}$ | | $1.1_{\pm 0.3}$ | | |
| KD | $50.4_{\pm 1.4}$ | $40.3_{\pm 0.9}$ | | $34.3_{\pm 1.1}$ | | |
| LP | $51.8_{\pm 0.6}$ | $37.1_{\pm 0.8}$ | | $28.0_{\pm 0.8}$ | | |
| Transformer OTFusion acts Uniform | $2.2_{\pm 0.4}$ | $1.2_{\pm 0.1}$ | | $1.0_{\pm 0.0}$ | | |
| Transformer OTFusion acts Conductance | $2.3_{\pm 0.4}$ | $1.0_{\pm 0.1}$ | | $1.0_{\pm 0.0}$ | | |
| Transformer OTFusion acts DeepLIFT | $2.3_{\pm 0.6}$ | $1.0_{\pm 0.0}$ | | $1.0_{\pm 0.0}$ | | |
| Transformer OTFusion wts Uniform | $3.9_{\pm 0.8}$ | $1.5_{\pm 0.4}$ | | $1.2_{\pm 0.3}$ | | |
| Transformer OTFusion wts Conductance | $4.4_{\pm 1.3}$ | $1.3_{\pm 0.3}$ | | $1.2_{\pm 0.3}$ | | |
| Transformer OTFusion wts DeepLIFT | $4.1_{\pm 1.3}$ | $1.2_{\pm 0.2}$ | | $1.1_{\pm 0.3}$ | | |
| HF Gradient Uniform (Ours) | $49.9_{\pm 1.1}$ | N/A | | N/A | | |
| HF Gradient Conductance (Ours) | $55.5_{\pm 1.3}$ | N/A | | N/A | | |
| HF Gradient DeepLIFT (Ours) | $\mathbf{55.5}_{\pm 0.8}$ | N/A | | N/A | | |
| KF Gradient Uniform (Ours) | $54.1_{\pm 1.1}$ | $43.1_{\pm 0.7}$ | | $36.9_{\pm 0.8}$ | | |
| KF Gradient Conductance (Ours) | $54.7_{\pm 1.2}$ | $\mathbf{43.5}_{\pm 0.5}$ | | $\mathbf{37.4}_{\pm 0.8}$ | | |
| KF Gradient DeepLIFT (Ours) | $54.6_{\pm 1.2}$ | $43.4_{\pm 0.5}$ | | $37.3_{\pm 1.1}$ | | |

Table 16: **Test accuracy** comparison when fusing ViT networks on CIFAR-100 trained on the **full dataset**. Results for 2-way fusion are averaged over 3 seeds, while results for 4-way fusion are shown only for a single seed. Fusion was performed with 5000 samples from the full dataset, except for activations-based Transformer OTFusion, where a subset of 200 samples was chosen, following (Imfeld et al., 2023). Fine-tuning was performed for 200 epochs with a learning rate of $3 \cdot 10^{-4}$ and a cosine annealing with warm restarts scheduler with a minimum learning rate of $10^{-6}$. During fine-tuning, the base models failed to improve. This table is complimentary to Table 4.

| Method | 2-way Zero-Shot | 2-way Finetuned | 4-way Zero-Shot | 4-way Finetuned |
|---|---|---|---|---|
| Individual Models | $\mathbf{73.9}_{\pm 0.2}$ $73.4_{\pm 0.3}$ | $73.5_{\pm 0.3}$ $73.0_{\pm 0.3}$ | $\mathbf{74.1}$, $73.6$, $73.0$, $72.9$ | $73.7$, $73.2$, $72.7$, $72.7$ |
| Ensemble | $75.7_{\pm 0.3}$ | $75.5_{\pm 0.4}$ | $76.6$ | $76.4$ |
| Vanilla Averaging | $1.9_{\pm 0.2}$ | — | $1.1$ | — |
| Transf. OTF acts Uniform | $2.7_{\pm 0.2}$ | $73.8_{\pm 0.4}$ | $1.0$ | $63.7$ |
| Transf. OTF acts Conductance | $2.4_{\pm 0.8}$ | $73.7_{\pm 0.4}$ | $1.0$ | $63.0$ |
| Transf. OTF acts DeepLIFT | $2.3_{\pm 0.9}$ | $74.0_{\pm 0.4}$ | $1.0$ | $62.6$ |
| Transf. OTF wts Uniform | $4.3_{\pm 0.2}$ | $74.0_{\pm 0.4}$ | $1.0$ | $72.6$ |
| Transf. OTF wts Conductance | $3.2_{\pm 1.1}$ | $73.8_{\pm 0.3}$ | $1.0$ | $68.6$ |
| Transf. OTF wts DeepLIFT | $3.2_{\pm 1.5}$ | $73.9_{\pm 0.3}$ | $1.0$ | $68.8$ |
| HF Gradient Uniform (Ours) | $57.0_{\pm 1.1}$ | $74.8_{\pm 0.4}$ | N/A | N/A |
| HF Gradient Conductance (Ours) | $58.6_{\pm 1.1}$ | $75.0_{\pm 0.5}$ | N/A | N/A |
| HF Gradient DeepLIFT (Ours) | $58.6_{\pm 1.3}$ | $75.2_{\pm 0.6}$ | N/A | N/A |
| KF Gradient Uniform (Ours) | $\mathbf{63.0}_{\pm 1.2}$ | $75.2_{\pm 0.5}$ | $57.5$ | $75.2$ |
| KF Gradient Conductance (Ours) | $62.8_{\pm 0.9}$ | $\mathbf{75.4}_{\pm 0.1}$ | $57.5$ | $75.2$ |
| KF Gradient DeepLIFT (Ours) | $62.4_{\pm 1.9}$ | $75.2_{\pm 0.1}$ | $57.1$ | $\mathbf{75.6}$ |

## G.2.2 TINY-IMAGENET

Table 17: **Test accuracy** comparison when fusing ViT networks on Tiny-ImageNet for **Sharded 2-way splits**. Fusion was performed using 5000 data points sampled from the dataset seen by the first model. For "activations-based" (i.e. acts) Transformer OTFusion, following (Imfeld et al., 2023), we used a subset of 200 samples. The weights-based variant (wts) does not use data.

| Method | 2-WAY SPLIT |
|---|---|
| Individual Model 0 | $28.3_{\pm 0.2}$ |
| Individual Model 1 | $27.5_{\pm 0.5}$ |
| Ensemble | $43.9_{\pm 0.4}$ |
| Vanilla Averaging | $0.9_{\pm 0.4}$ |
| KD | $31.0_{\pm 0.6}$ |
| LP | $33.2_{\pm 0.6}$ |
| Transformer OTFusion acts Uniform | $2.6_{\pm 1.2}$ |
| Transformer OTFusion acts Conductance | $2.7_{\pm 1.5}$ |
| Transformer OTFusion acts DeepLIFT | $2.6_{\pm 1.4}$ |
| Transformer OTFusion wts Uniform | $4.0_{\pm 0.9}$ |
| Transformer OTFusion wts Conductance | $2.3_{\pm 0.8}$ |
| Transformer OTFusion wts DeepLIFT | $1.9_{\pm 0.5}$ |
| HF Gradient Uniform (Ours) | $30.1_{\pm 2.1}$ |
| HF Gradient Conductance (Ours) | $32.9_{\pm 1.6}$ |
| HF Gradient DeepLIFT (Ours) | $\mathbf{33.5}_{\pm 1.3}$ |
| KF Gradient Uniform (Ours) | $32.3_{\pm 1.6}$ |
| KF Gradient Conductance (Ours) | $32.1_{\pm 1.4}$ |
| KF Gradient DeepLIFT (Ours) | $32.6_{\pm 2.0}$ |

Table 18: **Test accuracy** comparison when fusing ViT networks on Tiny-ImageNet trained on the **full dataset**. Results for 2-way fusion are averaged over 2 seeds. Fusion was performed with 5000 samples from the full dataset, except for activations-based Transformer OTFusion, where a subset of 200 samples was chosen, following (Imfeld et al., 2023). Fine-tuning was performed for 200 epochs with a learning rate of $10^{-5}$ and a cosine annealing with warm restarts scheduler with a minimum learning rate of $10^{-6}$. This table is complimentary to Table 5.

| Method | 2-WAY ZERO-SHOT | 2-WAY FINETUNED |
|---|---|---|
| Individual Models | $\mathbf{52.7}_{\pm 0.2}$ | $53.2_{\pm 0.5}$ |
| | $51.7_{\pm 0.0}$ | $51.7_{\pm 0.0}$ |
| Ensemble | $54.9_{\pm 0.4}$ | $55.7_{\pm 0.4}$ |
| Vanilla Averaging | $1.1_{\pm 0.1}$ | — |
| Transf. OTF acts Uniform | $1.4_{\pm 0.1}$ | $53.6_{\pm 0.1}$ |
| Transf. OTF acts Conductance | $1.4_{\pm 0.3}$ | $53.7_{\pm 0.2}$ |
| Transf. OTF acts DeepLIFT | $1.2_{\pm 0.3}$ | $53.7_{\pm 0.2}$ |
| Transf. OTF wts Uniform | $3.1_{\pm 0.2}$ | $53.8_{\pm 0.1}$ |
| Transf. OTF wts Conductance | $2.2_{\pm 0.8}$ | $53.6_{\pm 0.1}$ |
| Transf. OTF wts DeepLIFT | $2.1_{\pm 0.9}$ | $53.7_{\pm 0.3}$ |
| HF Gradient Uniform (Ours) | $40.5_{\pm 1.5}$ | $53.0_{\pm 0.3}$ |
| HF Gradient Conductance (Ours) | $42.1_{\pm 4.9}$ | $53.6_{\pm 0.2}$ |
| HF Gradient DeepLIFT (Ours) | $41.9_{\pm 3.2}$ | $53.7_{\pm 0.1}$ |
| KF Gradient Uniform (Ours) | $42.5_{\pm 0.5}$ | $53.9_{\pm 0.1}$ |
| KF Gradient Conductance (Ours) | $42.6_{\pm 0.8}$ | $\mathbf{54.2}_{\pm 0.4}$ |
| KF Gradient DeepLIFT (Ours) | $42.9_{\pm 0.3}$ | $53.8_{\pm 0.8}$ |

## H EXISTING ASSETS AND LICENSES

We make use of code from the following sources:

1. OTFusion Singh and Jaggi (2020), Open Source, https://github.com/sidak/otfusion.
2. ViT-CIFAR omihub777, MIT License, https://github.com/omihub777/ViT-CIFAR/blob/main/LICENSE.
3. Captum Kokhlikyan et al. (2020), BSD 3-Clause License, https://github.com/pytorch/captum/blob/master/LICENSE.

## I BROADER IMPACT

This work concerns foundational research on model fusion algorithms. We do not foresee any negative applications beyond those broadly applicable to model fusion algorithms.

As with other fusion methods, negative societal impacts may follow from using biased or harmful models as a base model to perform fusion as the fused model may contain the biases / harmful potential of the base model.

