# OpenReview forum: "Model Fusion via Neuron Interpolation"
_ICLR.cc/2026/Conference — ICLR 2026 Conference Withdrawn Submission_

### Official Review · Reviewer_MnPz · 2025-10-27

**Soundness:** 3
**Presentation:** 3
**Contribution:** 3
**Rating:** 4
**Confidence:** 4

**Summary:**

The process of model fusion is non-trivial, resulting the three key gaps in prior research, including reproducibility, base model quality, and heterogeneous data. Thus, this paper proposes a neuron-centric family of model fusion algorithms regardless of training data distribution as these algorithms incorporate neuron attribution scores into the fusion process.

**Strengths:**

1. It casts fusion as a principled representation matching problem, yielding a two-stage algorithm, which measure grouping error and approximation error. It decouples the traditional objective by introducing an auxiliary vector, enabling a more tractable decomposition of the cost function.
2. It incorporates neuron saliency into alignment, improving performance across our methods and enhancing existing approaches.
3. It provides a flexible opensource re-implementation of existing algorithms.

**Weaknesses:**

1. This method view DNN as a function parameterized by weights and for many model architectures, this function can be decomposed into many subfunctions. However, the rapid development of network architecture has made many SOTA network architectures no longer cascading, but containing many intricate connections. The authors do not discuss these complex network architectures.
2. Does a collection of pretrained base models to share the same network architecture, that is, do they need to be siamese networks? Is it possible for models with different network architectures to merge?
3. It lacks the comparison with SOTA model fusion methods, especially those proposed in the recent two years.

**Questions:**

1. For the three key gaps in prior research, a more intuitive explanation or illustration is needed as some gaps, especially the third gap, are professionally in-depth and difficult to understand.
2. In Eq. (1), $s_j$ is not introduced.
3. For data of different distributions, whether the auxiliary vector needs to be retrained as the optimal solution of the grouping error may be changed.

---

> ### Author Response · Authors · 2025-11-16
>
> We thank you for your thorough and constructive review. We will make sure to incorporate it into revisions and next versions of our paper.
>
> Below, we will address the weaknesses and questions you raised, to better explain our contribution.
>
> &nbsp;
>
> **Responses to Weaknesses**
>
> 1. Our method can be adapted to architectures with arbitrarily dense branches or skip connections, as long as it is possible to identify target “layers” to match between the base models. We have shown this for ViTs and ResNets which naturally involve skip branches. It is conceptually straightforward to adapt them for LSTMs or DenseNets, for example.
>
>      &nbsp;
>
> 2. This is possible. See, for example, Table 7, where we compress a ResNet34 into a ResNet18. The ResNet34 is trained on the full CIFAR-100 dataset and the ResNet18 is initialized by training on ⅓ of the data. The fusion is done only using a subset of the data the ResNet18 is given and the targets are computed only using the larger model’s activations. We also extend this to ResNet18 + ResNet50 and compare it with the same experiment for two ResNet34s (with one getting full data and the other ⅓ of the data).
>
>
>      **Extended Table 7: ResNet Compression for CIFAR-100, fusion dataset using ⅓ of the samples**
>      | Setting | Ensemble | Knowledge Distillation | KF-Gradient | Linear Probing | Model 0 | Model 1 |
>      |---------|----------|--------------|--------------------|----------------------|--------------|--------------------|
>      | 34 -> 18 | 50.44    | 30.98                   | 70.19                | 48.30                    | 29.18   | 82.35   |
>      | 50 -> 18 | 56.18    | 30.60                | 57.24                | 47.92                  | 29.18   | 83.35   |
>      | 34 -> 34 | 47.39    | 33.19             | 72.79                | 47.11                  | 29.41   | 82.35   |
>
>
>      We also provide an additional ablation study where we fuse a normal VGG16 with a VGG16 containing 0.5x, 1x, 2x, or 4x the parameters.The setting is a non-iid split on CIFAR-10 where each model gets half the data.
>
>
>      **Additional Table: Fusing a normal VGG16 with VGG16 with changed parameter counts, non-iid split on CIFAR-10**
>      | Fusion Setting       | Ensemble | Knowledge Distillation | KF-Gradient Conductance | KF-Gradient DeepLIFT | KF-Gradient Uniform | KF-Linear Conductance  | KF-Linear DeepLIFT | KF-Linear Uniform | Linear Probing | Model 0| Model 1| OTFusion Conductance | OTFusion DeepLIFT| OTFusion Uniform |
>      |---------------------|----------|--------------|--------------------|---------------------------|-----------------------|----------------------|------------------------|---------------------|--------------------|--------------|--------------------|---------|---------|------------------|
>      | With 0.5x | 88.92    | 84.68                  | 88.07                     | 87.89                 | 87.99                | 86.97                  | 86.57               | 85.74              | 88.19                | 87.07   | 72.23   | 34.35            | 38.18          | 56.57         |
>      | With 1x| 88.90    | 84.87                   | 88.03                     | 87.76                 | 87.78                | 86.39                  | 86.65               | 85.76              | 88.29                   | 87.07   | 74.41   | 36.40            | 37.75          | 50.63         |
>      | With 2x   | 89.38    | 84.56                 | 87.46                     | 87.83                 | 87.74                | 86.13                  | 86.35               | 85.08              | 88.22                 | 87.07   | 74.63   | 25.59            | 26.18          | 29.81         |
>      | With 4x  | 88.91    | 84.95               | 87.73                     | 87.77                 | 87.76                | 86.16                  | 86.25               | 84.92              | 87.89              | 87.07   | 74.93   | 15.95            | 15.08          | 18.59         |
>
>
>
>
> 3. We note that the Transformer OTFusion [1] baseline was published in ICLR 2024.
> While we agree that having more baselines is always preferable, we believe the selected baselines cover the standard methods in the literature with problem settings that are comparable to ours.

---

> ### Author Response · Authors · 2025-11-16
>
> **Responses to Questions**
> 1. We thank the reviewer for the critique and will review this in following revisions.
> It is introduced on line 147. “We also use sj for the importance score of neuron j (of the concatenated outputs z).”
>
>      &nbsp;
>
> 2. Our method is reliant on a fusion dataset for which the preactivations are computed to perform fusion. Since the base models are not retrained, this means their preactivations with regard to the fusion dataset remain fixed. Accordingly, this means that the auxiliary vector and grouping error can be fixed at the beginning of the fusion algorithm.
>
>      &nbsp;
>
> 3. With regards to different data distributions, here we always use a subset of the data provided to one of the base models, which is particularly important for the non-IID and sharded settings to simulate situations where the full training data cannot be shared. However, since we fix the fusion dataset the optimal solution does not change during the runtime of the algorithm.
>
> &nbsp;
>
> ---
>
> [1] Moritz Imfeld, Jacopo Graldi, Marco Giordano, Thomas Hofmann, Sotiris Anagnostidis, and Sidak Pal Singh. Transformer fusion with optimal transport. arXiv preprint arXiv:2310.05719, 2023.

---

> > ### Author Response · Authors · 2025-11-23
> > **Follow-up on Rebuttal**
> >
> > We apologize for the additional message -- we know you’re very busy. However, we believe our rebuttal has addressed the weaknesses and questions you raised in your review. We would greatly appreciate any follow-up comments you may have, and we remain happy to clarify anything further.

---

### Official Review · Reviewer_54JN · 2025-10-31

**Soundness:** 1
**Presentation:** 1
**Contribution:** 1
**Rating:** 2
**Confidence:** 4

**Summary:**

This paper proposes a new family of model fusion algorithms, termed "Neuron Interpolation," designed to merge multiple trained neural networks into a single representative model. The core idea is to frame fusion as a layer-by-layer representation-matching problem. The method operates in two stages per level: 1) a "grouping" step, which clusters the neuron outputs of all parent models (using K-means or Hungarian matching) to find importance-weighted "target" cluster centers , and 2) an "approximation" step, which trains the weights of the fused model's current level to match these target centers. The authors introduce two main variants: Hungarian Fusion (HF) for one-to-one matching of equal-sized models and K-means Fusion (KF) for the general case. A key contribution claim is the incorporation of neuron attribution (saliency) scores into this process. The paper presents experiments across various data distributions (full, non-IID, and "sharded") , claiming to significantly outperform prior methods like OTFusion and Git Re-Basin, especially in challenging zero-shot scenarios.

**Strengths:**

- **Problem Motivation:** The paper correctly identifies a significant and practical problem: existing fusion methods struggle in zero-shot and non-IID settings, which are common in real-world applications like Federated Learning.
- **Flexible Framework:** The proposed two-stage (grouping, fitting) framework is flexible. The K-means Fusion (KF) variant can naturally handle fusing models of different widths  (as shown in Table 1, if it were filled out), and the gradient-based variant can be applied to arbitrary differentiable layers

**Weaknesses:**

- **High Complexity and Sensitivity:** The authors' best method (gradient-based KF) is admitted to be "sensitive to hyperparameters" and its lack of robustness is shown by requiring two different settings for the paper's own experiments. It is also computationally expensive, running 16-23x slower than baselines (Table 10).
- **Overstated Saliency Contribution:** The claim of being the "first" to use saliency scores is factually incorrect, as the authors admit in Appendix C that prior work (Singh and Jaggi, 2020) already proposed it. Furthermore, the empirical gain from these scores is minimal (Tables 6, 7), contradicting the abstract's emphasis on this contribution.
- **Limited Novelty:** The method's novelty is limited, as it combines two well-known concepts: 1) neuron alignment via matching/clustering (as in OTFusion) and 2) multi-teacher feature-based distillation (which the authors call "fitting"). This combination is an incremental step, not a fundamental breakthrough.
- **Poor Presentation of Results:** The experimental section is poorly structured and difficult to follow. Critically, the authors often present tables of results without adequately discussing them or, in some cases, even not referencing them in the main text. This is a major oversight that hinders review.

**Questions:**

1. **Missing Related Work on Heterogeneity:** The related work section appears to miss some key references focused on heterogeneity. Could the authors elaborate on how their work, especially KF, differs from and compares to the cross-layer alignment method for heterogeneous networks in [1]? Furthermore, given the strong Federated Learning (FL) motivation, how does this approach relate to other federated methods designed to address client heterogeneity, such as [2]?

2. **Clarification of HF for Multi-Model Fusion:** The Hungarian Fusion (HF) method is described as solving a one-to-one matching problem, which is well-defined for the two-model case. However, the paper also presents results for multi-model fusion. How is the one-to-one matching problem formulated and solved when fusing $N > 2$ models?

3. **Addressing Error Propagation:** The paper claims a key weakness of prior work is "ignoring how the fused model evolves as the algorithm iterates through the levels... without accounting for potential changes [in] previous level outputs" (Lines 162-165). However, in the simple two-model case with linear levels and uniform importance, HF seems to reduce to a process very similar to OTFusion. Could the authors clarify the *exact* mechanism by which their method solves this alleged error propagation issue?

[1] Nguyen, Dang, et al. "On cross-layer alignment for model fusion of heterogeneous neural networks." ICASSP 2023-2023 IEEE International Conference on Acoustics, Speech and Signal Processing (ICASSP) . IEEE, 2023.

[2] Makhija, Disha, Nhat Ho, and Joydeep Ghosh. "Federated self-supervised learning for heterogeneous clients."arXiv preprint arXiv:2205.12493 (2022).

---

> ### Author Response · Authors · 2025-11-15
>
> We thank you for your thorough and constructive review. We will make sure to incorporate it into revisions and next versions of our paper.
>
> **Responses to Weaknesses**
>
> 1. We acknowledge that linear variants of our method are slower than previous works. However, this cost reflects the improved zero-shot performance of our representations-based fusion. We might compare this tradeoff to the case of the baseline fusion algorithms vs vanilla averaging which is obviously much faster yet generally ineffective. Moreover, our implementation is not optimized; significant speedups are feasible through a more efficient GPU-accelerated clustering, batched K-means, and parallel computation of target representations across layers. Optimizing these components is an engineering task rather than a conceptual limitation.
>
>     &nbsp;
>
>      Importantly, in **non-IID** and **sharded** regimes – where zero-shot fusion is fundamentally challenging – our method remains the only one among the evaluated baselines that achieves **meaningful improvements over the base models**, whereas prior approaches exhibit consistently poor zero-shot accuracy. Thus, although slower, our method provides a **practical solution to a setting for which no existing approach performs adequately.**
>
>     &nbsp;
>
>      In contrast, in full-dataset settings, all fusion methods (including ours and prior work) rely on a subsequent fine-tuning phase to recover performance and improve upon base models. In these cases, the runtime of this fine-tuning step dominates the overall cost, and the additional overhead introduced by our fusion procedure amounts to less than a 10% increase in the (total fusion + fine-tuning) runtime. Thus, the relative overhead is small in the scenarios where models are trained on the same data distributions.
>
>     &nbsp;
>
> 2. We have indeed written that it was first **“proposed”** by [3]. However, we would like to point out that saliency scores were never **“used”** by [3]. Indeed, importance scores are only mentioned in a single sentence in [3] proposing a method to incorporate them as future work without accompanying experimental results. We have produced experimental results for this proposal for the first time and shown it to be ineffective (see, for example, tables 12, 13, 14). Subsequent recent work applying OTFusion for fusion such as [5] also did not include experiments with saliency scores.
>
>      &nbsp;
>
>      While we agree that in some experiments the gains are minimal, in others (tables 12, 13, 15) they add significantly, particularly for our HF method in the sharded and non-iid settings which may motivate the incorporation of scores for the predominant align and average based fusion methods.
>
> &nbsp;
>
> 3. While our methods indeed build on ideas of neuron alignment and multi-teacher distillation, we argue that we have made several innovations:
>     &nbsp;
>
>     **a.** While we agree that the concept of neuron **alignment** is well-known, we disagree that neuron **clustering** in the context of model fusion is. Existing “well-known” neuron alignment methods [3, 4] primarily do so by computing a (soft) permutation matrix such that each neuron in one base model must be fused with a neuron from another model. While we provide HF for comparison with this restricted setting, we believe our KF algorithm to be the first fusion algorithm to allow for fusing multiple neurons in the same base model together.
>
>      If this is an oversight in our review of the literature, we would greatly appreciate it if the reviewer could share appropriate references showing that neuron clustering (as opposed to alignment) methods are well-known so that we might add the appropriate citations.
>
>     &nbsp;
>
>      **b.** Unlike existing alignment methods, which align and directly average the weights, we fit new weights for each level. This differs from multi-teacher distillation significantly in that the weights constructed are forced to minimise the error at each level instead of only minimizing the loss at the output. As we show empirically in Tables 3, 13, this allows the performance to exceed that of KD significantly in some cases.
>
>      Furthermore, we provide closed form solutions for the case of fully linear layers which is only possible when considering the layers one at a time in contrast with multi-teacher distillation.
>
>      In particular, this weight refitting step is where HF differs from OTFusion and compensates for error propagation as it allows the new weights to account for the actual activations of the fused model in the previous layers. Empirically, this results in a significant improvement in the performance of the fused model especially for zero-shot fusion.
>
>     &nbsp;
>
>      **c.** To our knowledge, this is the first work to integrate neuron importance (saliency) scores into the model fusion process, thereby revealing a principled connection between neuron interpretability and fusion, which are areas that have so far evolved separately.

---

> ### Author Response · Authors · 2025-11-15
>
> 4. We thank the reviewer for their critique. We will more clearly structure the experimental section in revisions.
> We have realized upon review that we neglected to cite table 7 in section 5.5.2. We apologize for this mistake, and we have fixed this in the revision.
>
> **Responses to Questions**
> 1. CLAFusion [1] primarily aims to fuse models with different numbers of layers. We refer to the following text from their abstract:
>
>      “While enjoying its success, OTFusion requires the input networks to have the same number of layers. To address this issue, we propose a novel model fusion framework, named CLAFusion, to fuse neural networks with a different number of layers, which we refer to as heterogeneous neural networks, via cross-layer alignment.”
>
>      &nbsp;
>
>       In that way, it can be viewed as an extension of OTFusion to work for base models with different numbers of layers. As they elaborate in section A of their appendix, OTFusion is used in their framework and thus inherits all the limitations of OTFusion except being able to fuse models with different numbers of layers. In particular, they explicitly state their method is unable to handle transformers.
>
>      &nbsp;
>
>       In this work, we mostly evaluated the case where the number of layers is the same for both models as we show in Tables 4,5 as it is the main scenario considered in the literature. Nonetheless, we also briefly consider the case with different numbers of layers in section 5.5, Table 7. Please also see our reply to Reviewer 4 for an extension of this experiment.
>
>      &nbsp;
>
>       In comparison with federated methods, which assume the ability to train the base models using their local data (for example, [2] which the reviewer has cited writes “In our framework, the learning on each client is also guided by peers to bring about collaboration.”), our method relies on a fusion dataset and in theory only requires the transfer of the base model’s preactivations on that dataset. Here the fusion dataset we use is a subset of the data used by a single base model, and thus our method can still be applied even if all other base models’ data are not available or cannot be shared.
>
>      &nbsp;
>
> 2. HF is not intended for fusion on more than 2 models. This is shown in Table 1, but we agree this may have been otherwise not well elaborated on in the text and will revise this. This was the motivation behind proposing KF, which can handle the case with more than 2 models.
>
>      &nbsp;
>
> 3. Please refer to our response to Weakness 3.b.
>
>      We agree this should have been better elaborated on in the text and will address this in further revisions.
>
> &nbsp;
>
> ---
>
> [1] Nguyen, Dang, et al. "On cross-layer alignment for model fusion of heterogeneous neural networks." ICASSP 2023-2023 IEEE International Conference on Acoustics, Speech and Signal Processing (ICASSP) . IEEE, 2023.
>
> [2] Makhija, Disha, Nhat Ho, and Joydeep Ghosh. "Federated self-supervised learning for heterogeneous clients."arXiv preprint arXiv:2205.12493 (2022).
>
> [3] Sidak Pal Singh and Martin Jaggi. Model fusion via optimal transport. Advances in Neural Information Processing Systems, 33:22045–22055, 2020.
>
> [4] Samuel K Ainsworth, Jonathan Hayase, and Siddhartha Srinivasa. Git re-basin: Merging models modulo permutation symmetries. arXiv preprint arXiv:2209.04836, 2022.
>
> [5] Moritz Imfeld, Jacopo Graldi, Marco Giordano, Thomas Hofmann, Sotiris Anagnostidis, and Sidak Pal Singh. Transformer fusion with optimal transport. arXiv preprint arXiv:2310.05719, 2023.

---

> > ### Author Response · Authors · 2025-11-23
> > **Follow-up on Rebuttal**
> >
> > We apologize for the additional message -- we know you’re very busy. However, we believe our rebuttal has addressed the weaknesses and questions you raised in your review. We would greatly appreciate any follow-up comments you may have, and we remain happy to clarify anything further.

---

### Official Review · Reviewer_SE4e · 2025-10-31

**Soundness:** 2
**Presentation:** 3
**Contribution:** 2
**Rating:** 2
**Confidence:** 5

**Summary:**

This paper proposes a neuron-level approach to model fusion that decomposes the fusion objective into two complementary components: (i) grouping neurons from multiple models according to similarity and importance, and (ii) fitting a fused network’s neurons to the cluster centroids derived from these groups. Two algorithmic variants are presented:

- Hungarian Fusion (HF): a one-to-one neuron matching method for models with identical widths, formulated as a linear sum assignment problem.
- K-means Fusion (KF): a general-width method that clusters neurons across models using importance-weighted K-means.

The method leverages saliency scores (Conductance and DeepLIFT) to guide both grouping and fitting steps, aiming to emphasize important neurons during fusion. Experiments are performed on CNN (VGG) and ViT architectures across IID, Non-IID, and sharded data settings, with and without fine-tuning, claiming consistent improvements over baseline fusion methods such as Git-Rebasin and OTFusion.

The paper also claims partial theoretical guarantees for the linear case and reports practical improvements on mid-scale datasets such as CIFAR-100 and Tiny-ImageNet.

**Strengths:**

- The decomposition of the fusion objective into grouping and approximation stages is a useful formalization. It reframes neuron matching as a clustering-and-refitting process, connecting prior permutation and alignment-based methods to a broader optimization viewpoint.
- Integrating saliency measures to inform neuron grouping and weighting adds a novel, biologically-inspired dimension to the fusion literature, which has largely focused on structural similarity rather than neuron importance.
- The modular structure could inspire broader frameworks for neuron-level model alignment and repair.
- The notion of importance-weighted neuron grouping could be extended to parameter-efficient transfer and federated learning settings.
- Even if limited by data reliance, the approach provides a pathway toward more interpretable fusion via neuron attribution.

**Weaknesses:**

- **Title clarity**: Since “interpolation” is a standard aggregation term in the model merging literature, consider renaming to reflect the neuron-centric mechanism or saliency usage.
- While the neuron clustering view is fresh, the method seems to **heavily build on Git-Rebasin’s activation matching** and other alignment-based model fusion works (e.g., _Ainsworth et al., Git Re-basin: Merging Models Effectively, 2023_). The contribution mainly lies in incorporating saliency and centroid fitting, rather than introducing an entirely new paradigm. Claims of generality to “arbitrary differentiable levels” are not supported, since the derivations rely on layer-wise alignment and assume comparable architectures.
- **Fusion-data dependence**: Both neuron grouping (activations) and weight refitting (non-linear levels) depend heavily on a “fusion dataset.” However, no ablation explores sensitivity to data quantity or distribution. This weakens claims of applicability to federated learning, where shared data are scarce.
- **Baselines**: Missing strong contemporary baselines (e.g., permutation+LS fusion, repair/rescaling methods, and data-free fusion). Ensembles outperform MIN in non-IID regimes.
- **Scalability**: Experiments are confined to VGG and ViT-small; no scaling to large models (e.g., ResNet-50, ViT-B/16) or multi-model setups (K>4). Runtime and memory comparisons lack depth.

**Questions:**

1.  How does performance vary as fusion data become more limited or non-IID? Can the approach be adapted for data-free or privacy-restricted settings?
2. Does the alternating grouping/fitting procedure converge empirically? Are there oscillations in neuron assignments or objective values?
3. How stable are the importance weights across different baselines and input samples? Does randomizing them degrade performance significantly?
4. Are gains mainly from refitting the classification head?

---

> ### Author Response · Authors · 2025-11-16
>
> We thank you for your thorough and constructive review. We will make sure to incorporate it into revisions and next versions of our paper. Below, we will address the weaknesses and questions you raised, to better explain our contribution.
>
> &nbsp;
>
> **Responses to Weaknesses**
>
> 1. We acknowledge that the term interpolation is used broadly in the merging literature. In our context, “neuron interpolation” refers to the following mechanism: at each level, the grouping step produces a set of centroid representations summarizing the key activations across base models, and the fitting step optimizes the current layer of the fused network (with preceding layers frozen) to approximate these centroids. Thus, the fused model effectively interpolates the activations produced by the base models at that layer.
>
> &nbsp;
>
> 2. Our method differs fundamentally from Git Re-basin [1] and related layer-wise alignment procedures. For KF, the only similarity with Git-Rebasin is that they aim to minimize the squared loss. Git Re-basin relies on permutation matrices while KF relies on clustering. Git Re-basin only averages the base models’ weights, while our algorithms directly compute new weights to minimize the cost function.
>
>     &nbsp;
>
>     Even considering only HF, there is a significant change compared to existing layer-wise alignment procedures. These approaches treat each layer independently: permutations for layer are computed based solely on the base models’ activations at that layer, without accounting for how earlier fused layers alter the fused model’s forward pass, besides tracking past permutation matrices. In contrast, our framework explicitly tracks how the fused network evolves across levels. At each layer, the clustering and fitting objectives evaluate how the current fused model, up to that layer, approximates the induced cluster targets. This induces cross-layer consistency and a dependency on the fused network’s internal representations, which is absent in Git Re-basin [1] or OT-Fusion [2].
>
>     &nbsp;
>
>     We also note that, in contrast to HF, activation-based Git Re-basin has only been demonstrated on models with fully connected and convolutional layers. In particular, it has not been demonstrated on ViT architectures, and its extension to arbitrary architectures is non-trivial.
>
>     &nbsp;
>
>     Regarding our claim of applicability to differentiable architectures: Eq. (3) defines an objective that is differentiable with respect to the parameters of the current layer, provided that the layer itself is differentiable. More precisely, our algorithms apply to architectures that can be decomposed into differentiable levels – a property satisfied by virtually all standard neural network architectures.
>
> &nbsp;
>
> 3. We believe both aspects are already covered in the paper.
>
>     a) *Ablations on data quantity.* Table 6 shows an ablation when the fusion dataset is varied.
>
>     b) *Ablations on data distribution.* The exploration of how our algorithms perform under different distributions has been done **throughout the whole paper, where we considered 3 data settings for our experiments.** This is also reflected in the paper *summary of your review*, where you correctly stated “Experiments are performed on [...] **across IID, Non-IID and Sharded data settings**”.
>
> &nbsp;
>
> 4. We clarify the following points:
>
>     &nbsp;
>
>     a) *Permutation + LS fusion.* While we’re not entirely sure which paper is referred to by “Permutation + LS fusion”; we assume that maybe you meant [5]. We note that while [5] has been cited in our paper, reproducing it [5] was unfortunately not possible within our experimental timeline. The model fusion literature lacks standardized, general-purpose implementations, and thus all algorithms need to be re-implemented from scratch. It can be easily seen thus that benchmarking against all new algorithms is not feasible. Hence, that’s why we chose Git Re-basin and OTFusion as our baselines, as they’re one of the most established works in this field. The same holds for the REPAIR [4] paper, which was also cited in the main text of our paper.
>
>     b) *Data-free fusion.* Our comparisons include weight-based Transformer OT-Fusion [3], which, to the best of our knowledge, represents the strongest practical data-free baseline for ViTs available to date (Tables 16 – 18).
>
>     c) *Ensembles vs. MIN.* Ensembles outperform fused models in most non-IID regimes, a well-known empirical fact. The motivation of model fusion is to create single models that approximate the ensemble (whose performance cost scales with the number of models). As [2] writes, “Prediction ensembling refers to keeping all the models and averaging their predictions (output layer scores), and thus reflects in a way the ideal (but unrealistic) performance that we can hope to achieve when fusing into a single model.”. We would appreciate clarification on the term “MIN,” as we’re unsure about what it refers to.

---

> > ### Author Response · Authors · 2025-11-16
> >
> > 5. We perform experiments with K > 4 models (Tables 2, 12, 13, 15). Regarding larger architectures: prior fusion papers -- Git Re-basin [1], OTFusion [2] and Transformer OT-Fusion [3] -- also restrict their analyses to limited large-scale experiments due to the computational burden:
> >
> >     &nbsp;
> >
> >     a) Git Re-basin reports only zero-shot results on ResNet-50/ImageNet-1k (for a single seed!), without fine-tuning
> >
> >     b) OTFusion does not consider datasets larger in scale than CIFAR-100
> >
> >     c) Transformer OTFusion evaluates a single ViT pair trained from a shared initialization (i.e. base models *not independently trained*).
> >
> >     &nbsp;
> >
> >     All works were accepted at top-tier venues despite these constraints. While we agree that larger-scale studies would strengthen the paper, we argue that our experimental coverage is comparable to, and in some respects more extensive than, prior work.
> >
> > &nbsp;
> >
> > **Responses to Questions**
> >
> > 1. We believe that these questions have been answered in Section 5.2, Section 6 of the main paper. In summary:
> >
> >     &nbsp;
> >
> >     a) Fusion data is inherently non-iid, as the fusion dataset is a subset of the data used by a base model for training. Since for non-iid settings, the base models have skewed data to begin with, the fusion dataset will be skewed too. This highlights the fact that our algorithms achieve effective knowledge transfer for various classes without actually needing to see samples belonging to those classes.
> >
> >     b) Adapting the method to privacy-restricted settings is natural; our non-IID/sharded experiments explicitly target this motivation. Data-free fusion remains open, but recent evidence (e.g., [5]) shows promise using publicly available proxy datasets, as noted in Section 6.
> >
> > &nbsp;
> >
> > 2. Empirically, the procedure converges reliably. In ViT experiments, while we allow up to 100 fitting epochs per layer, training typically converges by ~70 epochs or earlier.
> >
> > &nbsp;
> >
> > 3. Importance scores exhibit stable behavior with respect to the number of data points used to compute them: using 1k, 5k, or the full training set yields diminishing returns beyond 1k examples. We did not test random scores; however, uniform weights are a reasonable fallback when saliency cannot be estimated.
> >
> > &nbsp;
> >
> > 4. In Appendix F of our paper, we distinguish two different hyperparameter settings used for fusion of the gradient based algorithm. Setting 1, which was used for full-dataset and sharded setups, **alters the model significantly throughout**; thus providing gains from the whole change, not from the classification head. Setting 2, which was used only in some non-iid runs, mainly profits from changes in the classification head, and is similar to Linear Probing (LP). However, we would like to highlight that LP is a special case of our algorithm where we only fuse the classification head.
> >
> >     For the linear variants of our algorithms, the model changes drastically throughout, thus profiting from changes in the whole network, not only from the classification head.
> >
> > &nbsp;
> >
> > ---
> >
> > [1] Samuel K Ainsworth, Jonathan Hayase, and Siddhartha Srinivasa. Git re-basin: Merging models modulo permutation symmetries. arXiv preprint arXiv:2209.04836, 2022.
> >
> > [2] Sidak Pal Singh and Martin Jaggi. Model fusion via optimal transport. Advances in Neural Information Processing Systems, 33:22045–22055, 2020.
> >
> >
> > [3] Moritz Imfeld, Jacopo Graldi, Marco Giordano, Thomas Hofmann, Sotiris Anagnostidis, and Sidak Pal Singh. Transformer fusion with optimal transport. arXiv preprint arXiv:2310.05719, 2023.
> >
> >
> > [4] Keller Jordan, Hanie Sedghi, Olga Saukh, Rahim Entezari, and Behnam Neyshabur. Repair: Renormalizing permuted activations for interpolation repair. arXiv preprint arXiv:2211.08403, 2022.
> >
> >
> > [5] Anshul Nasery, Jonathan Hayase, Pang Wei Koh, and Sewoong Oh. Pleas-merging models with permutations and least squares. In Proceedings of the Computer Vision and Pattern Recognition Conference, pages 30493–30502, 2025.

---

> > > ### Author Response · Authors · 2025-11-23
> > > **Follow-up on Rebuttal**
> > >
> > > We apologize for the additional message -- we know you’re very busy. However, we believe our rebuttal has addressed the weaknesses and questions you raised in your review. We would greatly appreciate any follow-up comments you may have, and we remain happy to clarify anything further.

---

### Official Review · Reviewer_kkpb · 2025-11-01

**Soundness:** 4
**Presentation:** 3
**Contribution:** 3
**Rating:** 4
**Confidence:** 3

**Summary:**

The paper proposes a neuron-centric model fusion framework that casts fusion as a layer-wise representation matching problem. The method operates in two stages per level: (1) a "grouping" step that clusters concatenated neuron outputs from base models to define a set of target representations (cluster centers), and (2) an "approximation" step that optimizes the fused model's current level to match these targets. The objective function (Eq. 1) is novel in its explicit incorporation of neuron attribution scores (e.g., Conductance, DeepLIFT) to weight the importance of matching specific neurons. The authors introduce two main variants: Hungarian Fusion (HF) for 1:1 matching of equal-width models and K-means Fusion (KF) for the general case, with both linear and gradient-based optimization schemes for the approximation step.

**Strengths:**

- The proposed framework is intuitive, flexible, and moves beyond simple weight permutation (like OTFusion or Git Re-Basin ) by actively fitting a new sub-network to an intermediate target representation
- The incorporation of neuron attribution scores into the fusion objective is a novel contribution, theoretically allowing the process to prioritize salient features
- The empirical results are strong, especially in zero-shot and non-IID "sharded" scenarios (Table 2, 15), where the method (KF Gradient) succeeds while established baselines like OTFusion fail completely
- The ResNet compression experiment (Table 7) is a compelling demonstration of the method's strength, showing a >2x accuracy improvement over standard Knowledge Distillation using the same limited data

**Weaknesses:**

- The gradient-based variants, which produce the best results, are admittedly sensitive to hyperparameters. The paper provides two starkly different configurations (Setting 1 vs. Setting 2, Table 9)  without a clear ablation or principle for choosing between them. This significantly undermines the method's robustness and practicality
- The central novelty claim—incorporating attribution scores —is weakly supported by the main experimental results. For the flagship ViT experiments (Table 15, 16), the gains from using Conductance or DeepLIFT over uniform weights are consistently marginal or non-existent. For example, in Table 16 (fine-tuned), KF Gradient Uniform (75.2%) performs identically to DeepLIFT (75.2%) and is on par with Conductance (75.4%). The paper fails to analyze why the scores help significantly in some niche cases (VGGs, Table 13 ) but not in the main ViT results.
- In the challenging sharded setups (Table 2, 15), standard Knowledge Distillation (KD) and Linear Probing (LP) are surprisingly strong baselines that the paper does not sufficiently contextualize. For the 4-way ViT split (Table 15), KF Gradient (43.5%) is only marginally better than KD (40.3%). The authors state that their "Setting 2" (used for most non-IID models) is similar to LP, as most gains occur at the head. This suggests the complex, layer-wise grouping may be superfluous in these settings.
- The paper dismisses FedMA as a "non-zero-shot" baseline because it requires "retraining... after the alignment of every layer". However, the proposed "Gradient version of KF" also performs optimization (i.e., retraining) at every level via SGD. This distinction seems artificial, and the lack of comparison to FedMA is a clear omission.
- The runtime comparison (Table 10) shows the linear variants (HF/KF Linear) are 1-2 orders of magnitude slower (14x-83x) than OTFusion. This makes them practically unusable as fast alternatives, while the gradient-based methods are presumably even slower.

**Questions:**

- The empirical benefit of attribution scores is marginal in key ViT results (Table 15, 16). Can you provide a clear hypothesis for when these scores are beneficial and why they fail to provide significant gains for ViTs?
- Given that "Setting 2" (used for most non-IID models ) is described as degenerating to Linear Probing, does this imply the complex layer-wise grouping is unnecessary for these scenarios?
- How does the proposed gradient-based KF, which optimizes weights level-by-level, fundamentally differ in methodology and computational cost from FedMA, which you dismissed for its layer-wise retraining?
- The ResNet compression experiment (Table 7)  is strong, but it compares against standard KD. How does your method compare to more advanced, layer-wise distillation techniques?

---

> ### Author Response · Authors · 2025-11-16
>
> We thank you for your thorough and constructive review. We will make sure to incorporate it into revisions and next versions of our paper. Below, we will address the weaknesses and questions you raised, to better explain our contribution.
>
> &nbsp;
>
> **Responses to Weaknesses**
>
> 1. We agree that the gradient-based variants are sensitive to hyperparameter selection. As detailed in Appendix F.1, however, the hyperparameters were not tuned per model or dataset; instead, they depended almost exclusively on the **data regime**. This yields a simple actionable guideline:
>
>     a) Full-dataset and sharded setups --> Setting 1
>
>     b) Non-IID splits --> Setting 2
>
>     Thus, while sensitivity exists, the provided settings worked generally well in our experiments.
>
>     &nbsp;
>
> 2. We agree that the improvements from attribution-weighted fusion are smaller in the full-dataset ViT experiments. However, the central benefit of importance scores in our framework is primarily **zero-shot performance**, which is where fusion must operate without any retraining. Across all ViT zero-shot experiments (Tables 15-18), attribution scores consistently yield better fused models than uniform weighting, particularly in **non-IID or sharded settings**, where base models encode substantially different data distributions.
>
>     In contrast, in the full-dataset experiments where **fine-tuning is allowed**, the effect of importance weighting naturally becomes less evident: all weighting schemes provide “good enough’’ initializations from which fine-tuning can recover. In these settings, the performance of the fused model is dominated by the capacity of the subsequent fine-tuning stage, rather than by small differences in initialization.
>
>     &nbsp;
>
> 3. We apologize for the confusion. In the challenging **sharded** setup we employed **Hyperparameter Setting 1**, which does not reduce to LP and, in fact, produces fused models that lie substantially farther from the base models in weight space. As shown in Table 15, both HF and KF achieve **non-trivial gains over LP** (+3.7% and +2.9% respectively). Therefore, we do not share the view that layer-wise grouping might be superfluous in this setting.
>
>     &nbsp;
>
> 4. FedMA is methodologically distinct from our approach. FedMA repeatedly:
>
>     a) aligns models derived from a common initialization (previous federated communication round),
>
>     b) averages them to produce global parameters, and
>
>     c) redistributes the averaged model back to clients for further training on their local data.
>
>     d) repeats
>
>     &nbsp;
>
>     Our setting assumes a **single-shot fusion without access to the original training data**, and without the ability to repeatedly re-train full networks. This renders FedMA incompatible with our assumptions. Please refer to Section 5.2 of our main text for an explanation of our experimental setup. Additionally, reproducing FedMA to fit our experimental framework was beyond the scope of this paper.
>
>     Thus, the omission is due to fundamental methodological differences and reproducibility concerns, not an artificial distinction about “retraining.”
>
> &nbsp;
>
> 5. We acknowledge that linear variants of our method are slower than previous works. However, this cost reflects the improved zero-shot performance of our representations-based fusion. We might compare this tradeoff to the case of the baseline fusion algorithms vs vanilla averaging which is obviously much faster yet generally ineffective. Moreover, our implementation is not optimized; significant speedups are feasible through a more efficient GPU-accelerated clustering, batched K-means, and parallel computation of target representations across layers. Optimizing these components is an engineering task rather than a conceptual limitation.
>
>     &nbsp;
>
>     Importantly, in **non-IID and sharded** regimes -- where zero-shot fusion is fundamentally challenging -- our method remains the only one among the evaluated baselines that achieves **meaningful improvements over the base models**, whereas prior approaches exhibit consistently poor zero-shot accuracy. Thus, although slower, our method provides a **practical solution to a setting for which no existing approach performs adequately.**
>
>     &nbsp;
>
>     In contrast, in **full-dataset** settings, all fusion methods (including ours and prior work) rely on a subsequent **fine-tuning** phase to recover performance and improve upon base models. In these cases, the runtime of this fine-tuning step dominates the overall cost, and the additional overhead introduced by our fusion procedure amounts to **less than a 10%** increase in the (total fusion + fine-tuning) runtime. Thus, the relative overhead is small in the scenarios where models are trained on the same data distributions.

---

> > ### Author Response · Authors · 2025-11-16
> >
> > **Responses to Questions**
> >
> > &nbsp;
> >
> > 1. Attribution-weighted objectives are most useful when **models differ significantly in the distribution of data they were trained on** (non-IID or sharded regimes). In such cases, neuron importance varies across models, and weighting helps the fused network preserve salient subfunctions. When base models are trained on similar data (full-dataset scenario), their internal representations already overlap significantly (see [1]), making attribution weighting less influential.
> >
> >     Additionally, when fine-tuning is allowed, attribution weighting still provides a better starting point, often achieving peak accuracy in fewer fine-tuning epochs due to improved initialization.
> >
> > &nbsp;
> >
> > 2. In certain non-IID settings where the overlap between base-model distributions is extremely low, we indeed observed that LP barely outperforms our method under **Hyperparameter Setting 1**. Notably, both LP and our method under setting 1 outperform the individual base models in accuracy and loss. Because LP corresponds to a special case of our framework -- in which the first (n-1) levels are frozen and only the head is optimized -- we included it as one of the settings of our general algorithm. Thus, rather than suggesting that the layer-wise grouping is unnecessary, the results illustrate that our framework is flexible enough to interpolate between full fusion and LP-like behavior depending on the data regime. Furthermore, a simple cross-validation step on the fused model could help distinguish which setting performs best, in cases where the data regime of the base models is unknown.
> >
> >     However, we would like to point out that **in all other experiments** (full-dataset and sharded), **LP did not outperform our algorithms under Hyperparameter Setting 1**, which significantly alters the whole model, and not only the classification head. This can be seen in Tables 2, 3, 12, 13, 15.
> >
> > &nbsp;
> >
> > 3. Please see the response to Weakness #4. In summary: unlike FedMA, our method does **not** assume a shared initialization, client-server training cycles, or iterative global aggregation. We address the one-shot fusion problem, which FedMA does not.
> >
> > &nbsp;
> >
> > 4. The ResNet compression experiment is intended primarily as a *proof-of-concept* illustrating that our intermediate-representation matching objective facilitates more effective knowledge transfer than standard KD under skewed data constraints. Compression is not the main focus of our work, and we did not aim to benchmark against state-of-the-art layer-wise distillation methods, which often require specialized implementations or training protocols beyond the scope of our fusion setting. We view such comparisons as an interesting direction for future work.
> >
> > &nbsp;
> >
> > ---
> >
> > [1] Yixuan Li, Jason Yosinski, Jeff Clune, Hod Lipson, and John Hopcroft. Convergent learning: Do different neural networks learn the same representations? arXiv preprint arXiv:1511.07543, 2015

---

> > > ### Author Response · Authors · 2025-11-23
> > > **Follow-up on Rebuttal**
> > >
> > > We apologize for the additional message -- we know you’re very busy. However, we believe our rebuttal has addressed the weaknesses and questions you raised in your review. We would greatly appreciate any follow-up comments you may have, and we remain happy to clarify anything further.

---

### Note · Authors · 2025-12-02

**Comment:**

Due to the unusual circumstances during the rebuttal phase, we have withdrawn the submission. We appreciate the reviewers’ time and engagement.
For the benefit of future readers, we would like to highlight a few, non-exhaustive, clarifications that may help avoid potential misunderstandings when interpreting the paper:
- **Comparison with Federated Learning**: Our setting differs from most federated-learning approaches such as FedMA: Our methods do not assume a shared initialization, client-server communication rounds, or iterative global aggregation. In particular, our approach is designed for cases where only one of the base models’ training data is available for fusion, and where fusion can be performed only once, without recommunicating weights.

- **Neuron Importance Scores for Fusion**: To the best of our knowledge, our work provides the first experimental evaluation of using neuron-importance scores for model fusion. This idea was previously discussed in [1], but without empirical validation; in our experiments, we find that this strategy does not yield improvements for [1]. However, it yielded notable improvements for our methods, and [2] as well.

- **Fusion dataset experiments**: Our paper includes experiments concerning the fusion dataset. For all fusion runs, the fusion dataset is a subset of the “private” dataset of one of the base models. Since we consider different data distributions for the data seen by base models, our experiments illustrate performance when the fusion dataset distribution varies. These results are further supported by ablation studies (e.g., Section 5.2.2), where we used a skewed subset on a model trained on the full data.

- **Applicability to architectures with non-cascading connections**: Our methods are compatible with architectures that include residual or skip connections. This is demonstrated in the experiments section, where we present results for both ResNets and ViTs.

We hope these clarifications are useful to readers exploring related directions in the future.

[1] Sidak Pal Singh and Martin Jaggi. Model fusion via optimal transport. Advances in Neural Information Processing Systems, 33:22045–22055, 2020.

[2] Samuel K Ainsworth, Jonathan Hayase, and Siddhartha Srinivasa. Git re-basin: Merging models modulo permutation symmetries. arXiv preprint arXiv:2209.04836, 2022.

**Withdrawal Confirmation:**

I have read and agree with the venue's withdrawal policy on behalf of myself and my co-authors.